# BUILDARENA: A PHYSICS-ALIGNED INTERACTIVE BENCHMARK OF LLMS FOR ENGINEERING CONSTRUCTION

## ABSTRACT

Engineering construction automation aims to transform natural language specifications into physically viable structures, requiring complex integrated reasoning under strict physical constraints. While modern LLMs possess broad knowledge and strong reasoning capabilities that make them promising candidates for this domain, their construction competencies remain largely unevaluated. To address this gap, we introduce **BuildArena**, the first physics-aligned interactive benchmark designed for language-driven engineering construction. It takes a first step towards engineering automation using LLMs. Technically, it contributes to the community in two aspects: (1) an extendable task design strategy spanning static and dynamic mechanics across multiple difficulty tiers; (2) a 3D Spatial Geometric Computation Library for supporting construction based on language instructions. On eight frontier LLMs, **BuildArena** comprehensively evaluates their capabilities for language-driven and physics-grounded construction automation. We release the code here to benefit construction automation in engineering applications.

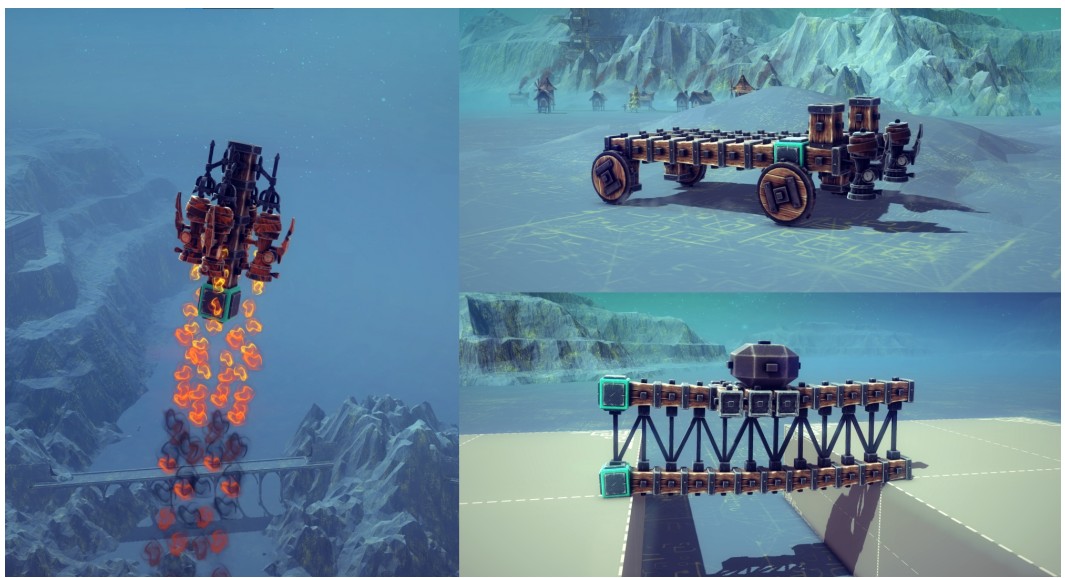

Figure 1: Examples of **BuildArena**'s construction results by LLMs, covering three tasks: **Lift** (left subfigure), **Transport** (upper right), and **Support** (lower right).

## 1 INTRODUCTION

Engineering construction automation is an important field of AI for Engineering. It has various applications in domains such as automotive, transportation, and civil infrastructure (Wang et al., 2005;

Domingues et al., 2016; Lin et al., 2019). The goal is to translate high-level task descriptions into executable end-to-end build plans that cover design, fabrication, and assembly. An ideal workflow lets users describe what they want in plain terms. For example, users request "*Design a rocket that meets Mars mission requirements*." The system then creates realistic parts with precise material details and manufacturing specs. The workflow also provides assembly instructions that can be integrated into production systems. Such automation capabilities promise significant improvements in engineering efficiency and productivity.

Language-driven automated construction presents challenges in two key aspects. On one hand, it necessitates physics simulation environments with high fidelity to real-world constraints so that virtual designs and assembly procedures adhere to geometric, physical, and structural constraints. While modern physics engines and robotics benchmarks offer robust simulation capabilities (Todorov et al., 2012; Coumans & Bai, 2016; Makoviychuk et al., 2021), there remains a gap in environments that integrate physics verification with language-driven multi-component assembly processes. On the other hand, the domain demands multilevel reasoning across long temporal horizons and 3D spatial contexts, as engineering artifacts inherently exhibit hierarchical organization, and their assembly follows sequential dependencies with strict feasibility constraints (Jiménez, 2013; De Fazio & Whitney, 2003). These factors collectively require both breadth of domain knowledge and depth of analytical thinking, challenging even for expert human engineers.

Large language models (LLMs) have progressed rapidly in recent years, accumulating broad world knowledge and demonstrating strong capabilities in language understanding, mathematical reasoning, and code generation (Brown et al., 2020; Guo et al., 2025; Shao et al., 2024; Roziere et al., 2023). Moreover, they have shown proficiency in following human instructions, generating plans, invoking tools, and composing executable programs to interact with the external environments (Yao et al., 2023; Schick et al., 2023; Qin et al., 2023). These general intelligent capabilities position LLM agentic systems as promising candidates for automatic engineering construction.

Despite these advances, current evaluations on LLMs provide insufficient evidence of their capacity to construct physical entities. Established LLM benchmarks predominantly assess mathematical and programming capabilities (Cobbe et al., 2021; Hendrycks et al., 2021; Chen et al., 2021), which are evaluated mainly in textual or static environments, without interactions with physical environments. Existing physical reasoning datasets focus on physics understanding, but neglect the multi-step construction processes (Bakhtin et al., 2019; Cherian et al., 2024). Meanwhile, research in programmatic 3D or CAD generation has advanced generation performance but rarely validates whether the generated designs yield executable assemblies under realistic physical conditions (Jones et al., 2020; Mallis et al., 2024). This interdisciplinary gap highlights the absence of frameworks to evaluate whether LLMs can effectively translate natural language specifications into physically viable assemblies. This limitation motivates our **research question**: *How can we comprehensively evaluate LLMs for language-driven and physics-grounded construction automation?*

In this paper, we answer the research question by proposing **BuildArena**, a physics-aligned interactive benchmark designed to assess LLMs' capabilities in engineering construction tasks. To our knowledge, this is the first benchmark that enables LLMs to perform 3D structure construction via natural language instructions and evaluates their performance within a physically constrained environment. **BuildArena** enables in-depth comparison and analysis of LLMs, featuring detailed logging. It by default consists of three components: task definition, LLM-based construction, and simulation-based evaluation. It supports customization of each component (see Figure 2). Examples of construction results are shown in Figure 1. Comparison between **BuildArena** and existing benchmarks are provided in Table 1 (*See Appendix B for more related work*), implying that our work takes a first step towards engineering automation using LLMs. Thus, it substantially expands the scope of current LLM benchmarks to 3D construction domains. Our technical contributions are summarized as follows.

- **We create an extendable task design strategy**. The strategy includes three task categories with quantifiable difficulty levels, along with corresponding evaluation metrics.
- **We develop a key framework module: a 3D Spatial Geometric Computation Library**. The module facilitates computations and feedback in iterative 3D construction to ensure accurate execution of LLMs' language instructions. As the geometric computation functions in the widely used Besiege simulator (Spiderling, 2018) are closed-source and inaccessible, our open-source module reproduces its building operations.

Table 1: Comparison between **BuildArena** and previous benchmarks.

| Benchmarks | Spatial Reasoning | 3D Construction | Construction-aimed Planning | Physical Simulator | Interactive Environment |
|---|:---:|:---:|:---:|:---:|:---:|
| PlanBench (Valmeekam et al., 2023) | ✗ | ✗ | ✗ | ✗ | ✗ |
| PlanQA (Rodionov et al., 2025) | ✓ | ✗ | ✗ | ✗ | ✗ |
| PHYRE (Bakhtin et al., 2019) | ✓ | ✗ | ✗ | ✓ | ✓ |
| VOYAGER (Wang et al., 2023a) | ✓ | ✗ | ✓ | ✓ | ✓ |
| Embodied Agent Interface (Li et al., 2024) | ✓ | ✓ | ✗ | ✓ | ✓ |
| **BuildArena (ours)** | ✓ | ✓ | ✓ | ✓ | ✓ |

## 2 METHOD

This section details our benchmarking methodology, which includes task setup in Section 2.1, language-driven and physics-grounded construction in Section 2.2, the LLM agentic workflow in Section 2.3, and the evaluation methodology in Section 2.4. Our method is illustrated in Figure 2.

### 2.1 TASK

To design tasks in a principled manner, we first abstracted a set of difficulty dimensions that are commonly encountered in engineering practice:

- **Quantification:** Extent of explicit numerical reasoning required (Kamble et al., 2024).

- **Robustness:** Tolerance to single-point failures (Zhao et al., 2023).

- **Magnitude:** Structural scale in span, load, and module count (Fan et al., 2023).

- **Compositionality:** Required depth of hierarchical substructure construction and integration (Thurairajah et al., 2023).

- **Precision:** Strictness of geometric requirement for placement and orientation (Gao et al., 2024b).

- **Ambiguity:** Clarity and completeness of task guidance (Moon et al., 2024).

Upon these dimensions, we construct three representative engineering task categories: **Support** (static structural stability), **Transport** (dynamic horizontal movement), and **Lift** (dynamic vertical movement). Within each task category, we defined three levels of difficulty, Easy (Lv.1), Medium (Lv.2), and Hard (Lv.3), by adjusting task details so that the corresponding requirements align with the above dimensions (see Figure 3). This design ensures both diversity of engineering scenarios (statics vs. dynamics, horizontal vs. vertical motion) and systematic coverage of engineering difficulty dimensions. Task descriptions, performance indicators and evaluation criteria of these three tasks are as follows. Detailed task content as LLM prompts are provided in Appendix G.

**Transport** focuses on constructing a mechanical structure capable of directional movement on a planar surface. It primarily examines the LLM agents' ability to exploit the spatial movement afforded by given building components. As the difficulty increases beyond Lv.1, the explicit instruction of building a four-wheeled vehicle is removed, and the transportation target changes from the machine itself into a cargo load with level-specific scale, adding challenges on both instruction interpretation and building larger structures. The **maximum transport distance** is chosen as the performance indicator, and we use a distance threshold as the criteria to identify if the machine is able to deliver effective transportation to the target.

**Support** requires constructing a static structure to support a load across a gap, aiming to test the ability of LLM agents to design and build bridges. The span of the gap multiplies across three levels, as larger span directly requires larger scale of the bridge, making the stable support harder as well. While only one is allowed in Lv.1, both Lv.2 and Lv.3 permit the modular construction with no more than three substructures without any detailed instruction, which also requires more precise assembly. We select **maximum load weight** as the performance indicator for bridges, and use a minimum threshold to determine if the bridge successfully supports the load.

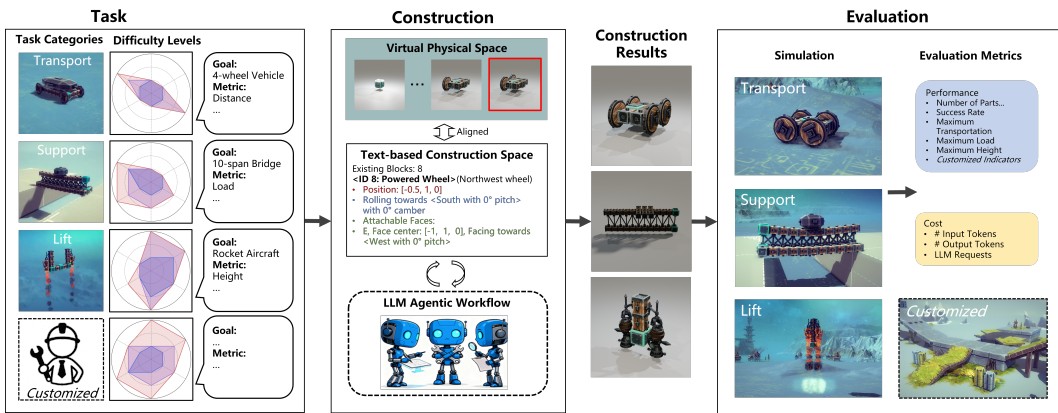

Figure 2: Illustration of our **BuildArena** framework. It contains three parts: (1) Task definition; (2) LLM-based Construction; (3) Simulation-based Evaluation. The arrows represent our pipeline. Components in dashed boxes, *i.e.*, task type, LLM agentic workflow, and simulator, could be customized by users. Details of the construction procedure is shown in Figure 4.

**Lift** requires constructing a rocket. At Lv.1, LLMs are explicitly required to build a single rocket engine without instruction on how to build, with **thrust-to-weight ratio** (TWR) as the performance indicator. TWR $> 1$ represents the feasibility of providing effective thrust and marks successful construction. At Lv.2, the task requires LLMs to construct a rocket-powered aircraft as an integrated single structure. At Lv.3, LLMs must first build two separate substructures (a rocket engine and a support frame) before assembling the two into an aircraft. Both Lv.2 and Lv.3 tasks require that the aircraft are capable of launching from the ground. **Maximum**

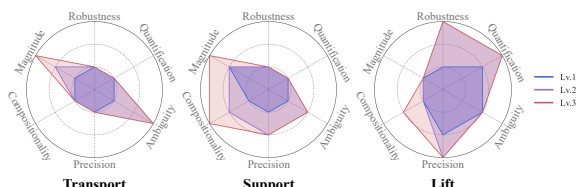

Figure 3: Difficulty profiles of the three task **Transport**, **Support**, and **Lift** across six engineering dimensions: Quantification, Redundancy, Scale, Modularity, Precision, and Ambiguity. Each radar chart illustrates how difficulty escalates from Lv.1 (blue) to Lv.2 (purple) and Lv.3 (red).

**Height** is adopted as the performance indicator, and the aircraft must reach a specific elevation to meet the success criterion. The escalation from Lv.1 to Lv.3 compounds multiple sources of engineering difficulty: higher demands on precise module alignment, the presence of multiple single points of failure (engine placement, structural balance), and strict requirements on modular construction and assembly. Together, these factors make **Lift** the most challenging task.

**Task Customization**. Tasks can be customized via textual prompts that define task constraints, objectives, testing procedures, and evaluation metrics. The prompts are fed into the LLM agentic workflow (see Section 2.3), which then executes the construction process.

## 2.2 LANGUAGE-DRIVEN AND PHYSICS-GROUNDED CONSTRUCTION

From the perspective of human engineering practice, construction is inherently an incremental and constraint-driven process. Structures are assembled step by step, each new component must connect to existing ones, and physical feasibility (e.g., collision avoidance) is continuously verified (Wilson & Latombe, 1994; De Fazio & Whitney, 2003). Each successful action requires accurate reasoning about the spatial relationships between new and existing structures. These features necessitate *Besiege* (Spiderling, 2018), an ideal platform to evaluate the LLMs for physics-grounded construction automation. Besiege is a popular construction sandbox game with realistic physics simulation, widely validated by world-wide player community to align with human physical intuition. It has a rich modules space, a complete collection of basic structural and functional module types that can

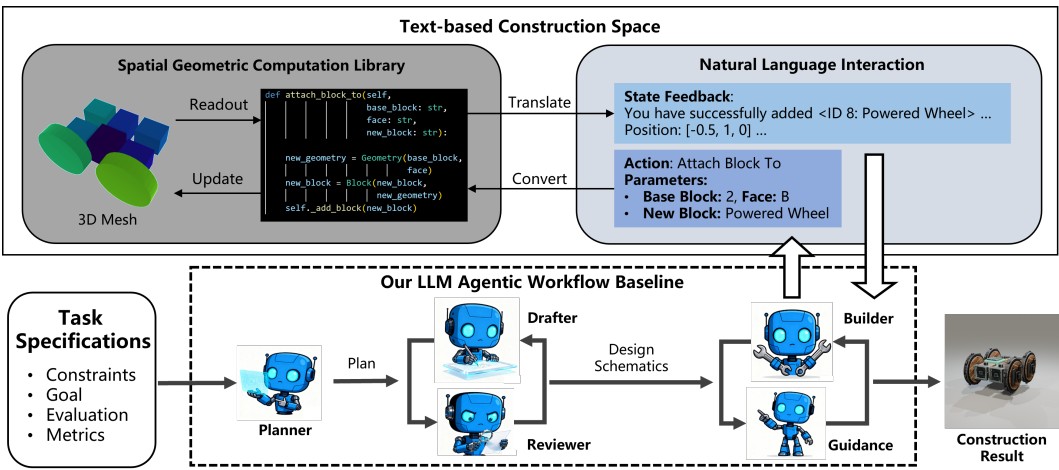

Figure 4: Details of the construction procedure in Figure 2. Our designed workflow (bottom row) contains five collaborative LLM entities, serves as a baseline for future user-customized alternatives. The text-based construction space (top row) has two transferable formats: code for physics-aligned spatial geometric computation, and natural language for LLM interface compatibility.

be combined to build complex objects, all completed by iteratively attachment and refinement of the native modules. Details about modules are provided in Appendix F.1.

However, Besiege only provides graphical representation of 3D structures in the interactive construction space for human users, instead of natural-language illustration for LLMs. It also only supports direct manipulation through physical controller inputs, without any interface for symbolic or language-based interaction, nor any indirect API for programmatically operating the construction process. Therefore, we develop an *open-source Spatial Geometric Computation Library that mirrors Besiege's closed-source construction logic and physical constraints*, enabling LLMs to interact with the construction space through language interfaces. It ensures consistency between the effects of actions executed by LLMs in the language space and those performed by human users in the graphic interface, as illustrated in Figure 4.

In implementation, it accepts an action announcing the operation and arguments from the LLM agent, computes and updates the state accordingly, and conducts physical constraint checks: it either returns a human-interpretable description of the current state, or prohibits the invalid action if certain constraints are violated and returns a prompt explaining the failure reason. All actions fall into four categories `Build`, `Refine`, `Query`, and `Control`, detailed in Appendix F.2.

## 2.3 LLM AGENTIC WORKFLOW

LLMs execute the construction procedure via LLM agentic workflow (Zhang et al., 2025a). For clear comparison between different LLMs, we restrict our consideration to workflows where all its entities employ the same LLM, differentiated by their respective prompts, as illustrated in Figure 2. After careful refinements, we finally obtained an effective workflow. Its design follows two principles: (1) a coarse-to-fine structure with an outline plan and progressing to granular, executable details (Xue et al., 2024); (2) a multi-party debate and multi-turn revision framework for incremental improvement of construction quality (Du et al., 2023). Based on these principles, our workflow employs five entities: `Planner` (P), `Drafter` (D), `Reviewer` (R), `Builder` (B), and `Guidance` (G). In addition, `Controller` (C) is used for the task **Transport**. Prompt details are provided in Appendix H. The workflow includes three stages, as shown in the bottom row of Figure 4:

- **Plan Phase**: Executed by `Planner`, this phase takes the task description and initial module set as input, outputting a structured construction plan in a predefined format.

- **Draft–Review Loop**: Based on the generated plan, `Drafter` produces design schematics. `Reviewer` reviews and verifies the schematics, guiding `Drafter`'s revisions. The loop repeats until approval; and terminates in failure if the plan violates predefined rules.

- **Build–Guidance Loop**: With approved schematics as input, `Builder` and `Guidance` collaborate on execution suggestions, building actions, and feedback. `Guidance` generates high-level suggestions step by step based on the draft, specifying the next action to invoke; `Builder` converts them into formatted construction commands for the Spatial Geometric Computation Library, which updates the states and returns either descriptive feedback or error prompts. The loop ends when `Guidance` confirms full completion, with the final state converted via the library into a conflict-free, simulation-compatible runnable file as the final result. Rejection by `Guidance` based on predefined rules terminates the process in failure.

This workflow serves as a baseline for the development of advanced workflows in the future. The workflow component can be substituted with any user-customized one.

## 2.4 EVALUATION

Our evaluation strategy is as follows. For each task-LLM pair, run the aforementioned construction procedure to produce a result (*e.g.*, a rocket) with detailed logs (*e.g.*, token consumption, conversation turns). This result is then placed in the simulation environment to operate, yielding evaluation metrics. To enhance reliability, the procedure is sampled 64 times for each task-LLM pair, with final reported results averaged over these 64 runs. Prompt and instructions for all tested LLMs are the same.

**Simulation environments are based on the Besiege sandbox**, with task-specific simulation protocols. For **Transport** tasks, the system evaluates if a constructed machine achieves effective motion. The LLM agent must specify a control configuration and sequence—invalid or missing controls cause immediate failure. The machine is then loaded into the environment. If repeated attempts show no effective movement, the system concludes the structure lacks mobility. For **Support** tasks, the environment provides fixed obstacles with varying gap widths according to difficulty level. A payload of gradually increasing weight is placed on the structure, and the simulation measures whether the machine can support and stabilize the load without collapse or loss of balance. For **Lift** tasks, Lv.1 records water cannons' heating status to calculate TWR; Lv.2–3 continuously activates water cannon firing to simulate launch, with module trajectories and cannons' heating status recorded for evaluation during a fixed window.

**Simulation environments can also be customized according to user-defined tasks**. Specifically, users set test conditions for the construction result—for instance, adding a load module above a constructed bridge—and configure simulation parameters, including the initial position, tracking points, and control information. All these configurations are implemented by invoking our developed Spatial Geometric Computation Library. Finally, **BuildArena** executes the simulation using a unified script and collects log data throughout the entire simulation process.

**Evaluation metrics cover performance and cost**. **Performance** includes three metrics: (1) Number of parts, referring to the count of modules present in the construction result. A smaller value is preferable because a core principle in engineering prioritizes simpler system structures to improve maintainability and reliability (El-Rayes & Khalafallah, 2005). (2) Success rate, defined as the proportion of trials that successfully passed the criteria among 64 samples, a higher value is better. (3) Performance indicator, a task-specific metric extracted from simulation data that evaluates the performance under realistic physical conditions. A higher value is preferable for all indicators. Detailed success criterion and indicator setup of each task are specified in Section 2.1. **Cost** is evaluated using three metrics: (1) number of accumulated input tokens, (2) number of output tokens, and (3) total number of LLM requests. A lower value is preferable for all the cost metrics.

## 3 EXPERIMENTS

In the experiments, we aim to answer the following two questions: (1) Whether **BuildArena** serves as an effective benchmark for testing the construction capabilities of LLMs? (2) How existing mainstream models perform within the **BuildArena** framework? To answer these questions, we evaluate eight closed-source LLMs in **BuildArena**, including GPT-4o, Claude-4, Grok-4, Gemini-2.0, DeepSeek-3.1, Qwen-3, Kimi-K2, and Seed-1.6. All simulations are conducted on Besiege. Model snapshots, module space, and simulation details are provided in Appendix E. We provide the **code** of **BuildArena** in this link.

Table 2: Average ($n = 64$) performance comparison on different tasks across task levels Lv.1 (easy), Lv.2 (medium), and Lv.3 (hard). The indicator means maximum displacement for **Transport**; maximum load for **Support**; TWR for Lv.1 and maximum height for Lv.2, Lv.3 of **Lift**. The best results are in **bold** and the second-best are underlined.

| Task | Model | Number of Parts | | | Success Rate (%)↑ | | | Indicator↑ | | |
|---|---|---|---|---|---|---|---|---|---|---|
| | | Lv.1 | Lv.2 | Lv.3 | Lv.1 | Lv.2 | Lv.3 | Lv.1 | Lv.2 | Lv.3 |
| **Transport** | GPT-4o | 11.0 | 13.1 | 18.4 | 9.4 | 1.6 | 7.8 | 13.5 | 4.2 | 5.1 |
| | Claude-4 | 9.5 | 13.4 | 26.1 | 17.2 | **4.7** | 15.6 | **34.9** | **6.8** | 7.3 |
| | Grok-4 | 12.0 | 15.6 | **34.6** | 25.0 | 0.0 | 9.4 | 19.6 | 4.2 | 12.3 |
| | Gemini-2.0 | 9.0 | 12.3 | 10.0 | 1.6 | 1.6 | 1.6 | 4.8 | 4.6 | 4.6 |
| | DeepSeek-3.1 | 9.2 | 11.6 | 16.7 | 6.2 | 0.0 | 1.6 | 6.2 | 4.6 | 4.6 |
| | Qwen-3 | 9.1 | 7.8 | 10.9 | 10.9 | 1.6 | 4.7 | 18.4 | 3.3 | **21.5** |
| | Kimi-K2 | **14.2** | **17.1** | 19.5 | 12.5 | 0.0 | 1.6 | 7.4 | 4.5 | 3.5 |
| | Seed-1.6 | 7.4 | 11.2 | 28.5 | 6.2 | **4.7** | 7.8 | 4.3 | 3.8 | 6.8 |
| **Support** | GPT-4o | **36.8** | 16.7 | 29.9 | 40.6 | 0.0 | 0.0 | 181.2 | 0.0 | 0.0 |
| | Claude-4 | 8.0 | 21.4 | 31.5 | 7.8 | 1.6 | 0.0 | 36.8 | 3.3 | 0.0 |
| | Grok-4 | 18.7 | 22.3 | 33.3 | 46.9 | 15.6 | 0.0 | **211.4** | **44.5** | 0.0 |
| | Gemini-2.0 | 20.5 | 23.5 | 41.0 | 23.4 | 0.0 | 0.0 | 105.5 | 0.0 | 0.0 |
| | DeepSeek-3.1 | 19.0 | 10.5 | 17.8 | 25.0 | 0.0 | 0.0 | 122.6 | 0.0 | 0.0 |
| | Qwen-3 | 18.6 | 18.0 | 22.2 | 12.5 | 4.7 | 0.0 | 70.5 | 13.9 | 0.0 |
| | Kimi-K2 | 23.3 | 34.7 | 16.5 | 29.7 | 4.7 | 0.0 | 122.6 | 18.6 | 0.0 |
| | Seed-1.6 | 33.4 | **36.2** | **68.8** | 45.3 | 9.4 | 3.1 | 197.4 | 25.8 | 7.1 |
| **Lift** | GPT-4o | 4.0 | 7.4 | 5.1 | 7.8 | 3.1 | 0.0 | 0.9 | 9.1 | 4.1 |
| | Claude-4 | 4.5 | 7.9 | 2.5 | 10.9 | 1.6 | 0.0 | 1.0 | 4.1 | 1.2 |
| | Grok-4 | 4.8 | 6.8 | 1.1 | **31.2** | **31.2** | 3.1 | **1.8** | **890.6** | **86.5** |
| | Gemini-2.0 | 4.3 | 6.3 | 4.9 | 0.0 | 0.0 | 0.0 | 0.5 | 2.8 | 0.8 |
| | DeepSeek-3.1 | 4.5 | 6.9 | 1.1 | 10.9 | 0.0 | 0.0 | 1.0 | 3.3 | 0.6 |
| | Qwen-3 | 3.5 | 7.2 | 3.2 | 3.1 | 0.0 | 0.0 | 0.6 | 2.7 | 0.8 |
| | Kimi-K2 | **5.3** | **15.1** | **12.0** | 6.2 | 3.1 | 0.0 | 0.7 | 44.8 | 1.8 |
| | Seed-1.6 | 3.5 | 3.3 | 0.0 | 6.2 | 0.0 | 0.0 | 0.9 | 1.7 | 0.0 |

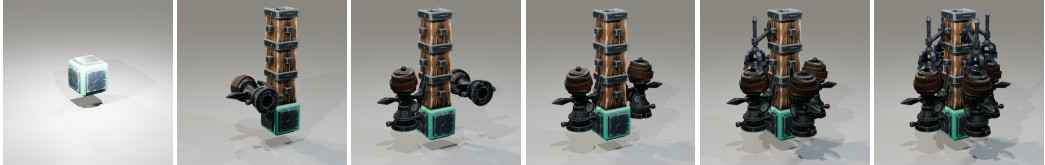

Figure 5: Example of the construction process. The rocket is constructed by Grok-4 for the **Lift** task under the Lv.2 (Medium) difficulty level. More examples are presented in Figure 10.

## 3.1 Effectiveness of **BuildArena**

The performance of eight models on **BuildArena** is presented in Table 2, with examples of construction results shown in Figure 9 and examples of construction procedures illustrated in Figure 5. These results demonstrate that, supported by the **BuildArena** evaluation framework, LLMs achieve language-based 3D construction automation, as evidenced by the following aspects. (1) Regarding task design, the diversity and difficulty levels are reasonably configured. Across individual tasks, performance tends to decrease as difficulty increases. An exception is the **Lift** task, where Lv.1 uses different metrics from Lv.2/3, making direct comparisons inappropriate. Specifically, at the Hard difficulty level of three tasks, most models exhibit low performance, yet a small number outperform others, indicating that the difficulty and criteria settings possess good discriminative power. (2) Concerning the LLM agentic workflow, numerous successful construction outcomes validate its effectiveness. This workflow enables collaborative behaviors among LLMs such as reflection (*e.g.*, the third subfigure from the left in Figure 5), which is essential for long-sequence planning. (3) Our Spatial Geometric Computation Library facilitates language-driven manipulation of the physical world. As illustrated in the construction procedure figures, these processes involve diverse actions including attachment, removal, rotation, shifting, and connection, which collectively meet the action requirements of construction tasks. (4) The simulator provides environmental sup-

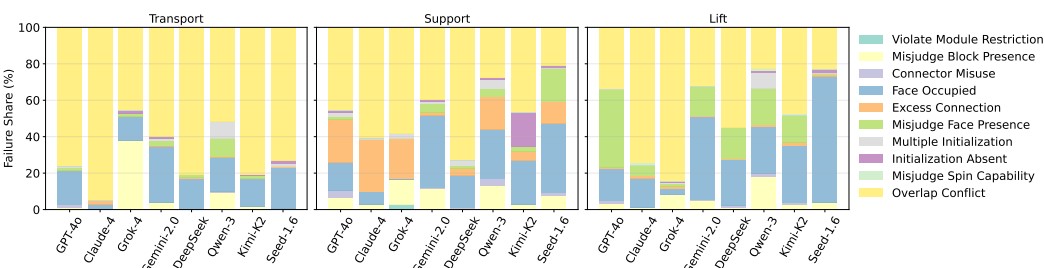

Figure 6: Distributions of failure reasons averaged over different LLMs.

port for the evaluation phase. For instance, it can place loads on bridges to test their load-bearing capacity and offer conditional support for the entire launch process of rockets. Overall, these key components of **BuildArena**, including task design, library, and simulator, collectively provide robust support for evaluation, enabling it to function as an effective and reliable benchmark. For ablation study on the multi-agent workflow, please refer to Table 4 of Appendix I. For more results about the decoding sensitivity, please refer to Table 5 of Appendix I. **BuildArena** can also be integrated into a closed-loop framework that uses feedback from simulator for iterative improvement. The details are presented in Table 6 of Appendix I.

## 3.2 PERFORMANCE OF LLMS

### 3.2.1 LIMITED PERFORMANCE OF CURRENT LLMS

From a complementary perspective, leveraging our **BuildArena** framework, current LLMs demonstrate elementary construction capabilities, as shown in Table 2. These capabilities are reflected in three aspects as follows. (1) in **Transport** tasks, as difficulty increases from Lv.2 to Lv.3—corresponding to an increase in payload size—all models adapt by scaling up the number of components to meet size constraints, thereby maintaining moving stability during simulation. Such patterns indicate that current LLMs effectively address challenges related to magnitude and ambiguity. (2) Second, when explicit constraints are relaxed, LLMs attempt unconventional solutions: they propose propulsion-powered carriers for **Transport** tasks and wheel-integrated bridge structures for **Support** tasks. In the latter case, LLMs explicitly state the need to utilize the automatic braking function of wheels to stabilize the bridge (see Figure 10, **Support** Lv.1 by DeepSeek-3.1). These behaviors highlight the potential of LLMs for creative exploration. (3) Remarkably, structures mirroring real-world engineering practices are constructed by LLMs, such as steel trusses in bridges and differential steering in vehicles (see Figure 10, **Support** Lv.2 by Grok-4). This suggests that structural concepts learned from text are not purely symbolic but carry implicit spatial information, enabling LLMs to instantiate them as feasible 3D structures. Notably, LLMs construct structures that align with real-world engineering practices, such as widely used steel trusses in bridges and differential steering systems in vehicles. This observation suggests that structural concepts learned from text are not purely symbolic but carry implicit spatial information, enabling LLMs to instantiate them as feasible 3D structures.

However, these models still suffer from significant limitations. (1) In hierarchical assembly tasks, such as the **Support** task, LLMs' success rates drop sharply as the assembly complexity, *i.e.*, the number of bridge substructures, increases. This indicates that the models ability to cope with compositional constructions is generally weak. (2) In high-precision tasks with low robustness, such as the **Lift** tasks, the model's success rate is generally extremely low. As the difficulty increases, most success rates drop to zero. This shows that existing models, with the exception of Grok-4, are unable to accomplish tasks that require high precision and suffer from strong sensitivity.

Figure 6 shows the occupation of different failure reasons during the construction process. Several features can be extracted from it. (1) **Spatial conflict is the most difficult mistake to avoid.** Overlap conflicts and attempts to use an already occupied face task the majority of failed actions. It indicates that LLM agents frequently fail to capture the updated spatial structures and make accurate next moves. (2) **Failure modes differ across task categories.** In the **Support** tasks, excess connection errors become more common, showing increasing attempts to reinforce the attachment. And in the

**Lift** tasks, more misjudge of face status emerges since the structures have less modules and thus less redundant faces for attachment.

### 3.2.2 COMPARISON AMONG DIFFERENT LLMS

The performance of different LLMs across six task difficulty dimensions is presented in Figure 7. It calculates the weighted score of each LLM across all the difficulty dimensions based on its score ranking in each task, followed by averaging the scores across the nine task-difficulty combinations. Key observations are as follows. (1) Grok-4 shows the strongest overall performance, which aligns with existing research (ARC Prize Foundation, 2025). In particular, Grok-4 exhibits exceptional performance in Precision and Robustness. (2) Aside from Grok-4, different models show a high degree of similarity in the distribution of their capabilities to handle tasks of varying difficulty. For each model, its strengths are consistently stretched in the Magnitude and Ambiguity dimensions, which is consistent with their perfor-

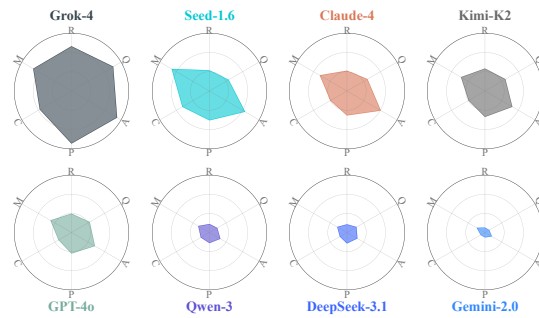

Figure 7: Performance of different LLMs against six dimensions of task difficulty: Quantification (Q), Robustness (R), Magnitude (M), Compositionality (C), Precision (P), Ambiguity (A).

mance in the **Transport** task in Table 2. In contrast, all LLMs exhibit consistent weaknesses across the other four dimensions. These findings provide clear directions for future improvements of LLMs.

### 3.2.3 COST ANALYSIS

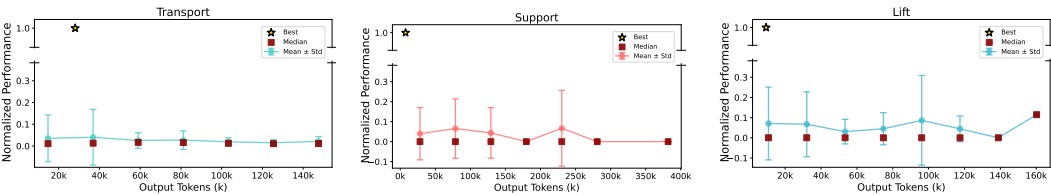

Figure 8: Trade-off between performance and cost. Longer output does not imply better results.

Figure 8 presents the relationship between cost and performance. Detailed results are provided in Appendix 3. From these results, we observation that **massive inference does not guarantee high performance.** In all the three tasks, best construction results often consume only moderate numbers of tokens, whereas many failed attempts incur massive token usage. This shows that beyond a certain capability threshold, additional inference cost does not translate into better performance.

## 4 CONCLUSION AND LIMITATIONS

In this work, we have introduced **BuildArena**, the first physics-aligned interactive benchmark designed to evaluate LLMs in engineering construction tasks. While our work represents the first step toward the promising domain of LLM-based engineering construction, it still has the following limitations. First, the framework lacks an extended outer loop to refine construction results based on simulator-derived evaluation outcomes, thereby failing to fully unlock the models' potential. Addressing this gap would involve designing an evaluation framework that enables closed-loop improvement driven by evaluation results. Second, the limited diversity of basic units in the module library constrains the range of constructible objects, a limitation that necessitates collaborative efforts from the research community to contribute a richer set of infrastructure assets. Looking forward, we believe our work paves the way for enabling and evaluating LLMs for complex engineering constructions, important for bridging LLMs with the physical world.

## REPRODUCIBILITY STATEMENT

The anonymous code is available at `https://anonymous.4open.science/r/BuildArena-9B7B/`. We provide a unified and modular code framework, together with scripts for reproducing all experiments.

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

## A  THE USE OF LARGE LANGUAGE MODELS (LLMs)

In this paper, the use of LLMs is limited to assisting in the writing and polishing of the paper. Additionally, we utilize Visual Languange Model (by Seed) to generate the cartoon images of the LLM agents shown in Figure 2 and Figure 4.

## B  RELATED WORK

**LLM Capability Benchmarks.** There have been numerous research evaluating the capabilities of LLMs recently, but few focus on construction tasks as **BuildArena**. In long-term planning, popular benchmarks such as PlanBench (Valmeekam et al., 2023), PlanGenLLMs (Wei et al., 2025), and PlanningArena (Zheng et al., 2025) all operate in abstract and static settings, ignoring physical constraints. Physical reasoning benchmarks like PhYRE (Bakhtin et al., 2019), PIQA (Bisk et al., 2020), Newton (Wang et al., 2023b), and ABench-Physics (Zhang et al., 2025b) do not involve assembly process. Spatial understanding work, such as GeoGramBench (Luo et al., 2025), PlanQA (Rodionov et al., 2025), reveals LLM blind spots in real-world layout reasoning but omits construction. None of them tie LLM capabilities to physics-constrained engineering assembly, whereas **BuildArena** uses a physics sandbox to evaluate interactive construction.

**Physics Simulation Environments.** Advanced engines (MuJoCo (Todorov et al., 2012), Isaac Gym (Makoviychuk et al., 2021)) and platforms (Autodesk (Autodesk Inc., 2024), SimScale (SimScale GmbH, 2024)) offer robust physics modeling, while digital twins create virtual structure representations (Dai et al., 2024). However, they lack integration with language interfaces for interactive construction, limiting their potential for following human instructions. LLM-based simulation studies focus on decision-making, not mechanical assembly (Kleiman et al., 2025; Gao et al., 2024a), and robot-centric environments (Gazebo, DART-LLM (Wang et al., 2024)) mainly target manipulation rather than structural building. **BuildArena** uniquely integrates a physics-aligned sandbox with a standardized language interaction protocol, tailored to evaluate the engineering construction driven by LLMs.

**AI-Driven Construction Automation.** AI applications in construction focus on parameterized design (Newton, 2019) and construction planning optimization (Zhang & Yang, 2025). Integrations with LLMs are still in the early stages(Ma12 et al.). The critical challenge of translating natural language to physically feasible structural assembly remains to be solved.

## C  MORE CONSTRUCTION RESULTS

More construction results are presented in Figure 9.

## D  MORE RESULTS OF CONSTRUCTION PROCEDURES

More results of construction procedures are presented in Figure 10.

## E  EXPERIMENTS DETAILS

### E.1  MODEL SNAPSHOTS

- Grok-4: grok4-0709
- GPT-4o: gpt-4o
- Claude-4: claude-sonnet-4-20250514
- Gemini-2.0: gemini-2.0-flash
- DeepSeek-3.1: deepseek-chat (DeepSeek-V3.1)
- Seed-1.6: doubao-seed-1-6-250615
- Kimi-K2: kimi-k2-turbo-preview

Table 3: Average ($n = 64$) cost comparison of models on different tasks across levels Lv.1 (easy), Lv.2 (medium), and Lv.3 (hard). The number of input/output tokens (# Input/Output Tokens) represents the cumulative total across multiple LLM requests required to complete one task instance on average.

| Task | Model | # Input Tokens ($\times 10^3$)↓ | | | # Output Tokens ($\times 10^3$)↓ | | | # LLM Requests↓ | | |
|---|---|---|---|---|---|---|---|---|---|---|
| | | Lv.1 | Lv.2 | Lv.3 | Lv.1 | Lv.2 | Lv.3 | Lv.1 | Lv.2 | Lv.3 |
| **Transport** | GPT-4o | 326.7 | 546.2 | 458.1 | 22.1 | 11.1 | 11.6 | 63.5 | 69.4 | 74.7 |
| | Claude-4 | 203.0 | 264.9 | 751.5 | 18.5 | 19.1 | 25.3 | 41.3 | 48.0 | 74.3 |
| | Grok-4 | 233.5 | 368.3 | 1259.9 | **8.9** | **10.8** | 15.5 | 40.7 | 49.3 | 98.4 |
| | Gemini-2.0 | 382.5 | 371.6 | **361.7** | 11.0 | **10.8** | **9.3** | 65.4 | 63.1 | **63.9** |
| | DeepSeek-3.1 | 252.8 | 469.4 | 715.5 | 18.5 | 23.6 | 28.2 | 49.8 | 67.9 | 80.1 |
| | Qwen-3 | 473.8 | 381.1 | 841.3 | 18.5 | 19.9 | 21.8 | 59.7 | 51.1 | 72.2 |
| | Kimi-K2 | 635.5 | 1099.6 | 968.7 | 13.9 | 16.0 | 14.2 | 82.2 | 96.8 | 99.8 |
| | Seed-1.6 | **197.1** | **248.7** | 899.3 | 41.0 | 43.2 | 63.7 | **34.1** | **41.1** | 81.5 |
| **Support** | GPT-4o | 1008.6 | 748.7 | 1671.8 | 19.5 | 19.0 | 22.5 | 102.5 | 135.7 | 192.5 |
| | Claude-4 | **116.3** | 548.5 | 1134.4 | 8.6 | 30.9 | 40.3 | **26.7** | 94.4 | 135.1 |
| | Grok-4 | 301.8 | 545.1 | 1152.0 | **7.6** | **11.9** | 14.7 | 45.5 | 69.4 | 91.1 |
| | Gemini-2.0 | 425.3 | 1413.9 | 2308.0 | 10.4 | 20.1 | 22.2 | 62.7 | 142.2 | 186.5 |
| | DeepSeek-3.1 | 484.4 | **299.0** | **424.8** | 22.1 | 19.3 | 18.4 | 70.5 | **57.0** | 64.8 |
| | Qwen-3 | 880.1 | 817.5 | 1263.5 | 17.7 | 23.9 | 23.3 | 63.8 | 89.4 | 104.0 |
| | Kimi-K2 | 508.8 | 1750.3 | 861.1 | 8.6 | 22.6 | **7.3** | 60.9 | 160.0 | **64.2** |
| | Seed-1.6 | 880.3 | 2043.2 | 5423.0 | 51.5 | 112.2 | 165.4 | 78.3 | 165.5 | 293.2 |
| **Lift** | GPT-4o | 345.7 | 232.3 | 423.6 | 23.7 | 8.7 | 13.8 | 51.2 | 50.8 | 88.1 |
| | Claude-4 | 279.8 | 376.0 | 386.8 | 22.6 | 28.8 | 30.6 | 44.9 | 50.7 | 68.9 |
| | Grok-4 | **103.4** | **180.9** | **128.0** | **6.7** | **8.4** | **10.2** | **24.2** | **33.0** | **34.3** |
| | Gemini-2.0 | 290.5 | 266.1 | 445.1 | 7.1 | 10.3 | 14.1 | 40.4 | 47.6 | 80.3 |
| | DeepSeek-3.1 | 317.3 | 401.6 | 396.8 | 22.5 | 24.7 | 29.5 | 54.4 | 61.3 | 76.0 |
| | Qwen-3 | 483.7 | 987.0 | 877.2 | 22.0 | 28.6 | 18.1 | 49.7 | 76.5 | 76.5 |
| | Kimi-K2 | 288.8 | 885.8 | 715.2 | 7.4 | 11.1 | 13.4 | 46.5 | 84.6 | 96.5 |
| | Seed-1.6 | 227.4 | 233.3 | 244.3 | 51.4 | 52.1 | 62.9 | 35.0 | 35.8 | 50.3 |

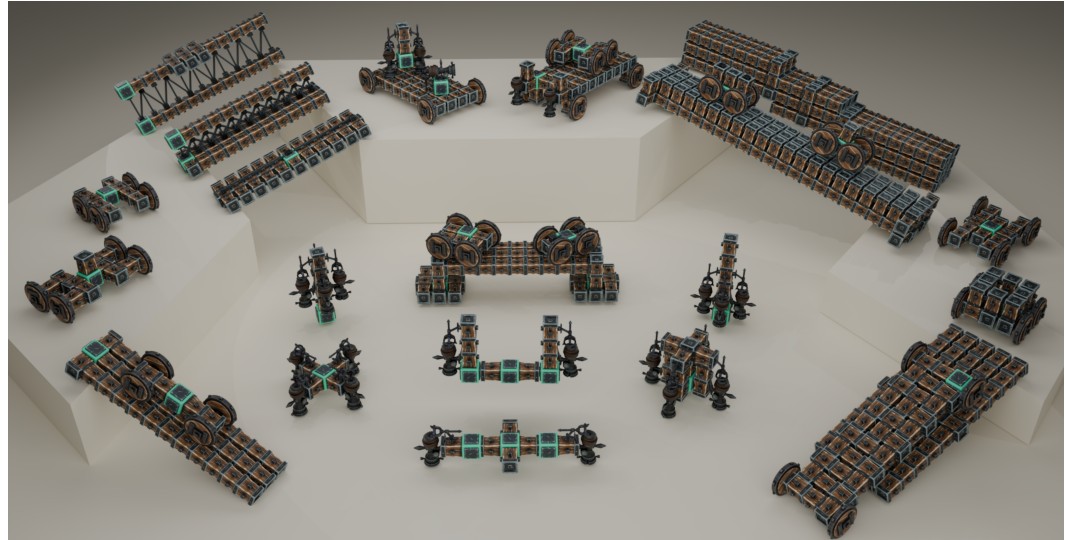

Figure 9: More examples of construction results of **BuildArena**, spanning three tasks: **Transport** (vehicles), **Support** (bridges) and **Lift** (rockets with nozzles).

• Qwen-3: qwen-plus (Qwen3 series)

All models are accessed via official API endpoints encapsulated by AutoGen (Wu et al., 2023) framework.

### E.2    BASIC MODULES

For our experiments, we have defined a set of six basic modules that serve as the fundamental building blocks for all tasks. This curated selection—comprising the small wooden block, powered wheel, water cannon, torch, brace, and winch—is sufficient to realize the necessary constructions without introducing overpowered or overly specialized parts. The detailed descriptions and specifications of each module are presented below.

#### E.2.1    4 KINDS OF AVAILABLE BLOCKS

**Powered Wheel (shape: [2, 2, 0.5], mass: 1.0)**

```
Description: A powered wooden wheel (diameter = 2, thickness = 0.5)
    rotates at a constant speed of 100 rpm, and automatically brakes when
     the wheel stops.  Each powered wheel can be individually controlled
    to rotate forward or backward by pressing and holding configurable
    control keys. The wheel's motion is governed by the following
    constraints: - The wheel's rotation axis is perpendicular to the
    attached face. - The rolling direction is always parallel to the
    attached face. - For example, if the attached face is a horizontal
    face, the wheel will also be horizontal; if the attached face is a
    vertical face, the wheel will also be vertical. - For example, if the
     wheel is attached to a side face, the wheel will be rotating
    parallel to the side face and the rolling direction is perpendicular
    to the side face. - For example, if the wheel is attached to a bottom
     face, the wheel will be rotating parallel to the bottom face and
    unable to roll effectively.
```

**Small Wooden Block (shape: [1, 1, 1], mass: 0.3)**

```
Description: Small wooden cubic block with shape of [1, 1, 1]
```

**Torch (shape: [1.5, 0.5, 0.5], mass: 1)**

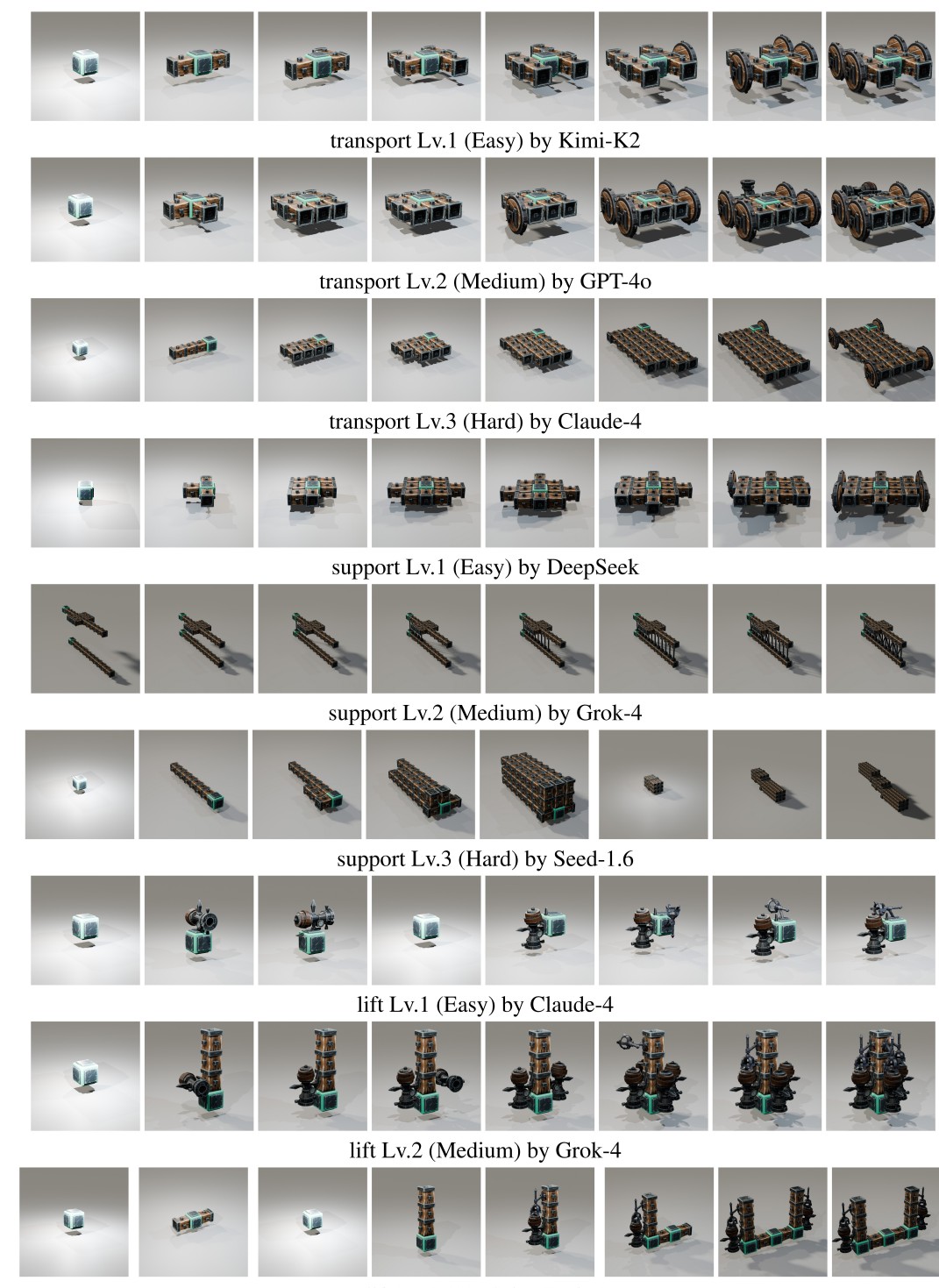

transport Lv.1 (Easy) by Kimi-K2

transport Lv.2 (Medium) by GPT-4o

transport Lv.3 (Hard) by Claude-4

support Lv.1 (Easy) by DeepSeek

support Lv.2 (Medium) by Grok-4

support Lv.3 (Hard) by Seed-1.6

lift Lv.1 (Easy) by Claude-4

lift Lv.2 (Medium) by Grok-4

lift Lv.3 (Hard) by Grok-4

Figure 10: Building procedure examples across 9 tasks.

```
Description: The torch flame sets flammable blocks, structures, and
    entities on fire.  It can be extinguished by Water Cannons (and Steam
    Cannons!), and reignited by other burning blocks. Their most common
    use is in heating water cannons so they produce steam, particularly
    in vanilla builds  However, they can be extinguished by steam plumes,
```

```
      so care must be taken not to fly backwards.  The torch will generate
      a spherical heating area with a radius of 0.3 unit in front of its
      flame nozzle direction (that is, the position of the torch body plus
      the orientation vector).  All objects in this area will be heated or
      ignited by the flame. They can also be used for setting fire to
      things for destructive purposes. Torches have no attachable faces for
       further attachment or connection. The Torch is shaped as a short
      horizontal support (length of 0.5), and a vertical shaft (length of
      1), the flame is at the end of the vertical shaft. For example, if
      the torch is attached to a vertical face (face center is [0.5, 0, 0])
       and points upwards, the torch coordinates will be [1, 0, 0] since
      the horizontal support of the torch has a offset of 0.5 from the
      attached surface, and the heating area will be a sphere with radius
      0.3 centered at [1, 0, 1] since the length of the vertical shaft is
      1.
```

**Water Cannon (shape: [2, 2, 1], mass: 1.5)**

```
Description: The Water Cannon sprays water in a fixed direction, which is
    determined by the attachment and orientation of the water cannon.
    Generates constant recoil force of 1.6 units of mass at normal
    gravity. The recoil force is not affected by speed or external
    conditions. Each water cannon can be individually controlled to fire
    by pressing and holding a configurable control key. Water Cannon has
    no attachable faces for further attachment or connection. Steam Mode:
     If any part of the water cannon is heated, it will fire steam
    instead of water and deliver 8.6 times the regular recoil force.
    Water Cannon has a peanut-shaped body (narrower in the middle than at
     the two ends, inlet and outlet are at the two ends) with length of
    1.75, width of 1, and height of 1. The middle part of the water
    cannon is narrower, making it hard to be directly heated if the heat
    source is small. For example, if the water cannon is attached to a
    vertical face (face center is [0.5, 0, 0]) and points downwards, the
    water cannon center coordinates will be [1, 0, 0] since the
    connection part of the water cannon has a offset of 0.5 from the
    surface, and the water cannon inlet will be at [1, 0, 0.75] and the
    water cannon outlet will be at [1, 0, -1] with a shape of 1.75x1x1
    cylinder (narrower in the middle than at the two ends).
```

### E.2.2    2 KINDS OF AVAILABLE CONNECTORS:

**Brace (mass: 0.5)**

```
Description: The Brace is a block that can be used to connect two
    separated blocks with a solid hinge. It can be used to enhance two
    blocks that are already connected together, or to assemble structures
     that are separated in the space. The mass of this block is always
    the same regardless of the length. Brace must be connected between
    two attachable faces of existing blocks, it cannot be directly
    attached to a single block.
```

**Winch (mass: 0.4)**

```
Description: The Simple Rope + Winch (simply as Winch or Rope) is a
    machine block composed of two winches at its end node which connects
    two blocks by a variable-length rope. Winch must be connected between
     two attachable faces of existing blocks, it cannot be directly
    attached to a single block.
```

### E.3    SIMULATION DETAILS

All simulations are conducted on Besiege v1.75 (build 23370) with Lua Scripting Mod (for controlling and logging), using the Steam distribution on Windows, performed in the native physics

settings of the game and executed by a unified automation script. Motion trajectories are recorded at a sampling rate of 25 Hz for subsequent quantitative analysis.

# F    3D SPATIAL GEOMETRIC COMPUTATION LIBRARY

## F.1    MODULE SPACE

The modules space $\mathcal{V}$ is a complete collection of basic module types like small wooden block, powered wheel, water cannon, torch, brace and winch that can be combined to build complex objects. The state in the construction procedures is represented as a triple $S = \langle V, \mathcal{P}, c \rangle$. Here, $V \subset \mathcal{V}$ denotes the set of modules involved in the current structure; The projection operator $\mathcal{P} : V \to \mathrm{SE}(3)$ maps each module in $V$ element-wisely to its 3D pose; and $c = \langle (t_1, k_1, \Delta t_1), \dots, (t_n, k_m, \Delta t_n) \rangle$ forms a control sequence, where $t_n$ is the timestamp of pressing control key $k_m$, $\Delta t_n$ is the press duration, and overlapping key-press operations are permitted.

To be more specific, each module $v \in \mathcal{V}$ is characterized by four attributes, expressed as $v = \langle \mathcal{F}, G, \gamma, \pi \rangle$. Geometric attribute $G$ encodes mesh, collision shape, initial orientation, and relative coordinates/orientations of connectable faces; $\mathcal{F}$ is a finite set of connectable faces where with each face is associated with a normal vector derived from $\mathcal{P}(v)$ and $G$, described via natural-language-aligned terms like letter labels and angular coordinates; $\gamma$ represents physical/functional parameters such as mass, rotational speed, thrust; control attributes $\pi$ maps control keys to actions and spin directions. Additionally, each module is accompanied by a natural-language summary $c_{\mathrm{init}}(G, \gamma, \pi)$ for initial prompts, with few-shot examples for modules (e.g., Powered Wheel) to help LLM agents align with critical functional info, such as rolling/jetting directions.

## F.2    ACTION SPACE

The action space comprises five categories of operators that cover the whole construction process: `Build`, `Refine`, `Assemble`, `Control`, and `Query`. A sequence of these actions forms a trajectory $\bar{a} = \langle (a_1, r_1), (a_2, r_2), \dots, (a_T, r_T) \rangle$, where each $a_i \in \mathcal{A}$ and $r_i$ denotes return information, including success / failure codes and natural-language descriptions of spatial structures. These categories are defined as follows:

`Build`: Enables core construction operations in the simulated environment, including module attachment, connection, removal, resetting, rotation, translation, and reversal.

`Refine`: When the structural state generated in the Build phase contains rotating modules, a subsequent Refine phase follows to allow fine-tuning of the structural state, in an attempt to ensure that the rotating modules have a reasonable rotation direction.

`Assemble`: If multiple substructures were built, an Assembly phase is then initialized, allowing the reuse of former construction results as building components.

`Control`: Manages control-related functionalities, such as updating action-to-control-key mappings in $\pi_v$ and appending control operations $(t, k, \Delta t)$ to the control sequence $\delta$ within $s$.

`Query`: Retrieves natural-language descriptions of structural states, including: a summary of the overall state $s$ as $r_t = c_s(\mathcal{V}, \mathcal{P})$; detailed module $v$ information as $r_t = c_v(\mathcal{F}_v, p_v)$; function-to-key mappings for control-enabled modules in $s$ as $r_t = c_s(\pi_1, \pi_2, \dots, \pi_v)$; descriptions of $s$'s control sequence as $r_t = c_s(\delta)$; and function-to-pose mappings for control-enabled modules in $s$ as $r_t = c_s(p_1, p_2, \dots, p_v)$.

## F.3    SPATIAL GEOMETRIC COMPUTATION LIBRARY

### F.3.1    BESIEGE SIMULATOR

Besiege is a physics-based construction sandbox game environment that enables the assembly and simulation of mechanical structures using modular components. It features a realistic physics engine, validated through extensive community use, which aligns closely with real-world physical principles. The environment includes a diverse set of structural and functional modules (over 70 types), allowing for the iterative construction of complex objects such as vehicles and static sup-

ports. These can be tested in simulated scenarios, with support for multiplayer validation and access to over 200,000 user-generated designs via an integrated workshop. In this work, we leverage Besiege as a neutral platform for evaluating language-driven construction under physical constraints, emphasizing its modular building system and simulation fidelity.

Besiege serves as the underlying simulation backend, providing a validated physics engine for testing constructed structures. As a construction sandbox, it simulates realistic dynamics, including gravity, friction, and module interactions, using Unity's physics system. Our library interfaces indirectly with Besiege by replicating its core operations (e.g., module attachment and control sequencing) without direct API access.

### F.3.2 SPATIAL GEOMETRIC COMPUTATION LIBRARY

The Spatial Geometric Computation Library implements the core functionalities for managing 3D spatial operations in the BuildArena framework. It handles state updates, geometric transformations, and constraint validations, bridging LLM-generated instructions to Besiege's physics simulation. Functions are organized into tool groups for modular use: **control** for sequencing actions, **build** for assembling structures, **refine** for post-attachment adjustments, **default** for querying states, and **build_only** for initialization. Below, we highlight representative functions from each group, with illustrations.

**Control Tool Group** This group manages timed control sequences for powered modules, enabling dynamic behaviors in simulated environments.

*add_control_sequence*: Facilitates the addition of timed inputs to simulate machine operations, essential for tasks requiring sequential activation.

```python
def add_control_sequence(time: float, key: str, hold_for: float) -> str:
    """Add a new control sequence entry."""
    # Implementation: Append to sequence list with validation
    return "Sequence added successfully."
```

*review_control_config*: Provides visibility into current control mappings, supporting iterative debugging during construction.

```python
def review_control_config() -> str:
    """A tool to review the current control configuration."""
    # Implementation: Aggregate and format control data
    return "Control config: [list of keys and actions]"
```

**Build Tool Group** This group supports the core assembly of structures, including attachments and connections under geometric constraints.

*attach_block_to*: Enables precise module placement on existing structures, enforcing face-based alignment for stable builds.

```python
def attach_block_to(base_block: Union[str, int], face: str, new_block:
    str, note: str) -> str:
    """Attach a new block to a face of an existing block."""
    # Implementation: Compute pose, check collisions, update state
    return "Block attached successfully."
```

*connect_blocks*: Establishes reinforced links between modules, crucial for enhancing structural integrity in complex designs.

```python
def connect_blocks(block_a: Union[str, int], face_a: str, block_b: Union[
    str, int], face_b: str, connector: str, note: str) -> str:
    """Connect two blocks using a connector."""
    # Implementation: Validate distance, add connector module
    return "Blocks connected successfully."
```

**Refine Tool Group** This group allows fine-tuning of module positions and orientations after initial placement, aiding in overlap resolution.

*twist_block*: Adjusts rotational alignment to optimize functional orientations, particularly for directional components.

```python
def twist_block(block_id: Union[str, int], angle: float) -> str:
    """Twist a block clockwise relative to its rooted surface."""
    # Implementation: Apply rotation matrix, update pose
    return "Block twisted successfully."
```

**Default Tool Group** This group provides state inspection tools, ensuring accurate feedback for LLM reasoning loops.

*get_machine_summary*: Offers a high-level overview of the current build state, mandatory for final validations before simulation.

```python
def get_machine_summary() -> str:
    """Get the latest state of the machine without face captions."""
    # Implementation: Summarize blocks and poses
    return "Machine summary: [overview details]"
```

**Build-Only Tool Group** This group handles initialization, setting the foundation for new constructions.

*start*: Initializes the build environment with a starting module, incorporating initial offsets for custom positioning.

```python
def start(init_shift: List[float], init_rotation: List[float], note: str)
    -> str:
    """Start to build the machine by creating and positioning the
        starting block."""
    # Implementation: Create initial block, apply transformations
    return "Starting block positioned."
```

## G  TASK DETAILS

Prompts of all the three tasks are listed as follows.

### G.1  TRANSPORT

#### G.1.1  EASY (Lv.1)

```
**Constraints:**
- Use only one sub-structure.
- The vehicle must have at least four wheels.
- The vehicle must be capable of forward driving and demonstrate a
    steering mechanism.
- Conventional steering mechanisms (e.g., rotating front wheels relative
    to the body) are not available with the provided blocks. Alternative
    steering strategies must be employed.

**Goal:**
- Drive the vehicle from the starting position (x=0, y=0) on the ground
    to the target position (x=10, y=10) on the ground (north-east
    direction) in the simulation environment.

**Evaluation Protocol:**
- The vehicle will be placed at (x=0, y=0) on the ground in the
    simulation environment.
```

```
- An open-loop control sequence will be programmed by a specialized AI
    agent following your plan, consisting of a list of commands with the
    format:
- [time: when to press the control key, command: the control key to press
    , duration: how long to hold the key]
- The trajectory of the vehicle will be recorded as feedback and
    optimized over three trials by adjusting the control sequence.
- The final score will be the best score across the three trials.

**Scoring Metrics:**
- *Trajectory Deviation:* Distance between the actual trajectory and the
    ideal straight-line path from start to target (smaller is better).
- *Structure Stability:* Whether the vehicle remains intact during
    driving (higher stability is better).
- *Time Efficiency:* Time taken to reach the target position (shorter is
    better).
- *Cost:* Number of blocks used to construct the vehicle (fewer is better
    ).
```

### G.1.2 MEDIUM (Lv.2)

```
**Constraints:**
- Use only one sub-structure.
- The cargo will not show in the building process, do not include it in
    the building plan.

**Goal:**
- Move a 2.5 × 2.5 × 1.5 cargo with 50 units mass from the starting
    position (x=0, y=0) on the ground to the target position (x=10, y=10)
     on the ground (north-east direction) in the simulation environment.

**Evaluation Protocol:**
- The machine will be placed at (x=0, y=0) on the ground in the
    simulation environment.
- The cargo will be loaded to the machine by freely dropping from above
    the starting position (x=0, y=0, z=3.5).
- The cargo will not have solid connection with the machine.
- An open-loop control sequence will be programmed by a specialized AI
    agent following your plan, consisting of a list of commands with the
    format:
- [time: when to press the control key, command: the control key to press
    , duration: how long to hold the key]
- The trajectory of both cargo and machine will be recorded as feedback
    and optimized over three trials by adjusting the control sequence.
- The final score will be the best score across the three trials.

**Scoring Metrics:**
- *Trajectory Deviation:* Distance between the actual trajectory of the
    cargo and the ideal straight-line path from start to target (smaller
    is better).
- *Structure Stability:* Whether the machine remains intact during
    driving (higher stability is better).
- *Time Efficiency:* Time taken to reach the target position (shorter is
    better).
- *Cost:* Number of blocks used to construct the machine (fewer is better
    ).
```

### G.1.3 HARD (Lv.3)

```
**Constraints:**
- Use only one sub-structure.
- The cargo will not show in the building process, do not include it in
    the building plan.
```

```
**Goal:**
- Move a 4 × 8 × 1.5 cargo (long axis along the north-south direction)
    with 50 units mass from the starting position (x=0, y=0) on the
    ground to the target position (x=10, y=10) on the ground (north-east
    direction), and back to the starting position in the simulation
    environment.

**Evaluation Protocol:**
- The machine will be placed at (x=0, y=0) on the ground in the
    simulation environment.
- The cargo will be loaded to the machine by freely dropping from above
    the starting position (x=0, y=0, z=3.5).
- The cargo will not have solid connection with the machine.
- An open-loop control sequence will be programmed by a specialized AI
    agent following your plan, consisting of a list of commands with the
    format:
- [time: when to press the control key, command: the control key to press
    , duration: how long to hold the key]
- The trajectory of both cargo and machine will be recorded as feedback
    and optimized over three trials by adjusting the control sequence.
- The final score will be the best score across the three trials.

**Scoring Metrics:**
- *Trajectory Deviation:* Distance between the actual trajectory of the
    cargo and the ideal straight-line path from start to target (smaller
    is better).
- *Structure Stability:* Whether the machine remains intact during
    driving (higher stability is better).
- *Time Efficiency:* Time taken to reach the target position (shorter is
    better).
- *Cost:* Number of blocks used to construct the machine (fewer is better
    ).
```

## G.2 SUPPORT

### G.2.1 EASY (Lv.1)

```
**Constraints:**
- Use only one sub-structure.

**Goal:**
- Build a bridge capable of spanning a gap between two flat terrains (5
    units wide, 5 units high).
- The bridge must be able to support a 2.5 × 2.5 × 1.5 cargo placed at
    its center.

**Evaluation Protocol:**
- The terrains are positioned with edges at (x=0, y=2.5, z=5) and (x=0, y
    =-2.5, z=5), forming a 5-unit-wide gap along the north-south axis
    with a vertical drop of 5 units.
- The bridge will be initially placed at (x=0, y=0, z=7), slightly above
    the terrain tops, so it can gently fall into position.
- There will be no fixed connection between the bridge and the terrain.
- A cargo of size 2.5 × 2.5 × 1.5 will be dropped at (x=0, y=0, z=7),
    directly above the center of the gap.
- The cargo will rest on the bridge without any fixed connection.
- The cargo's weight will gradually and linearly increase from zero (no
    initial impact).
- The trajectory of the cargo will be tracked; the load at which the
    cargo sinks below the gap will be recorded as the bridge's maximum
    supported load.
- If the bridge fails to span the gap or misses the cargo at the start,
    the score is 0.
```

```
**Scoring Metrics:**
- *Maximum Load:* Maximum load supported before the cargo falls below the
    gap (higher is better).
- *Cost:* Number of blocks used to build the bridge (fewer is better).
```

### G.2.2   MEDIUM (LV.2)

```
**Constraints:**
- Use no more than 3 sub-structures.

**Goal:**
- Build a bridge capable of spanning a gap between two flat terrains (10
    units wide, 5 units high).
- The bridge must be able to support a 2.5 × 2.5 × 1.5 cargo placed at
    its center.

**Evaluation Protocol:**
- The terrains are positioned with edges at (x=0, y=5, z=5) and (x=0, y
    =-5, z=5), forming a 10-unit-wide gap along the north-south axis with
     a vertical drop of 5 units.
- The bridge will be initially placed at (x=0, y=0, z=7), slightly above
    the terrain tops, so it can gently fall into position.
- There will be no fixed connection between the bridge and the terrain.
- A cargo of size 2.5 × 2.5 × 1.5 will be dropped at (x=0, y=0, z=7),
    directly above the center of the gap.
- The cargo will rest on the bridge without any fixed connection.
- The cargo's weight will gradually and linearly increase from zero (no
    initial impact).
- The trajectory of the cargo will be tracked; the load at which the
    cargo sinks below the gap will be recorded as the bridge's maximum
    supported load.
- If the bridge fails to span the gap or misses the cargo at the start,
    the score is 0.

**Scoring Metrics:**
- *Maximum Load:* Maximum load supported before the cargo falls below the
    gap (higher is better).
- *Cost:* Number of blocks used to build the bridge (fewer is better).
```

### G.2.3   HARD (LV.3)

```
**Constraints:**
- Use no more than 3 sub-structures.

**Goal:**
- Build a bridge capable of spanning a gap between two flat terrains (20
    units wide, 5 units high).
- The bridge must be able to support a 2.5 × 2.5 × 1.5 cargo placed at
    its center.

**Evaluation Protocol:**
- The terrains are positioned with edges at (x=0, y=10, z=5) and (x=0, y
    =-10, z=5), forming a 20-unit-wide gap along the north-south axis
    with a vertical drop of 5 units.
- The bridge will be initially placed at (x=0, y=0, z=7), slightly above
    the terrain tops, so it can gently fall into position.
- There will be no fixed connection between the bridge and the terrain.
- A cargo of size 2.5 × 2.5 × 1.5 will be dropped at (x=0, y=0, z=7),
    directly above the center of the gap.
- The cargo will rest on the bridge without any fixed connection.
- The cargo's weight will gradually and linearly increase from zero (no
    initial impact).
```

```
- The trajectory of the cargo will be tracked; the load at which the
    cargo sinks below the gap will be recorded as the bridge's maximum
    supported load.
- If the bridge fails to span the gap or misses the cargo at the start,
    the score is 0.

**Scoring Metrics:**
- *Maximum Load:* Maximum load supported before the cargo falls below the
    gap (higher is better).
- *Cost:* Number of blocks used to build the bridge (fewer is better).
```

## G.3 LIFT

### G.3.1 EASY (LV.1)

```
**Constraints:**
- Use only one sub-structure.

**Goal:**
- Build a single rocket engine capable of providing propulsion to a
    single direction.

**Evaluation Protocol:**
- The rocket engine will be placed at position (x=0, y=0, z=0) on the
    ground plane.
- During the simulation, the firing control key of the rocket engine will
    be pressed and held continuously.
- The vertical propulsion force of the rocket engine will be calculated
    by the difference in vertical position of the rocket engine between
    the start and end of the simulation.

**Scoring Metrics:**
- *Maximum Propulsion Force:* The maximum propulsion force achieved by
    the rocket engine (higher is better).
- *Cost:* The total number of blocks used to construct the rocket engine
    (fewer is better).
```

### G.3.2 MEDIUM (LV.2)

```
**Constraints:**
- Use only one sub-structure.

**Goal:**
- Build a rocket capable of lifting off from the ground and ascending
    into the sky in the simulation environment.

**Evaluation Protocol:**
- The rocket will be placed at position (x=0, y=0, z=0) on the ground
    plane.
- During the simulation, the firing control key of the rocket engine will
    be pressed and held continuously.
- The motion trajectory of the rocket will be recorded throughout the
    simulation.

**Scoring Metrics:**
- *Maximum Height:* The highest vertical position (z) reached by any
    block of the rocket (higher is better).
- *Trajectory Deviation:* The average lateral distance between the rocket
    's actual trajectory and the ideal vertical line (smaller is better).
- *Maximum Speed:* The highest speed achieved by any block of the rocket
    (higher is better).
- *Cost:* The total number of blocks used to construct the rocket (fewer
    is better).
```

### G.3.3 HARD (Lv.3)

```
**Constraints:**
- Use only two sub-structures.

**Goal:**
- Build a single rocket engine capable of providing propulsion to a
    single direction.
- Build a simple chassis to assemble the rocket engines using braces to
    form a symmetric rocket.
- The assembled rocket should be able to lift off from the ground to the
    sky in the simulation environment.

**Evaluation Protocol:**
- The assembled rocket will be placed at position (x=0, y=0, z=0) on the
    ground plane.
- During the simulation, the firing control key of the rocket engine will
     be pressed and held continuously.
- The motion trajectory of the assembled rocket will be recorded
    throughout the simulation.

**Scoring Metrics:**
- *Maximum Height:* The highest vertical position (z) reached by any
    block of the assembled rocket (higher is better).
- *Trajectory Deviation:* The average lateral distance between the
    assembled rocket's actual trajectory and the ideal vertical line (
    smaller is better).
- *Maximum Speed:* The highest speed achieved by any block of the
    assembled rocket (higher is better).
- *Cost:* The total number of blocks used to construct the assembled
    rocket (fewer is better).
```

## H WORKFLOW AND PROMPTS

Prompts of entities in the workflow are listed as follows. `Planner`:

```
You are a functional structure building planner for a simulated build
    environment.
Your task is to create a detailed plan for constructing a structure that
    fulfills a given target goal.
You will be provided with a goal and a list of available building blocks.
Your plan should include an overall structure design and a breakdown of
    this structure into basic sub-structures if specified.
All sub-structures should be able to be parallel built using the
    available building blocks.

Here are the available building blocks you can use:
<available_blocks>
{available_blks}
</available_blocks>

- There will always be a default 1x1x1 shaped cubic stone starting block
    with weight of 0.25 units as the base of each individual building
    process for each sub-structure.
- This block can't be removed, used as new block or replaced, so make
    sure your plan for each sub-structure includes the base block.

- The global coordinates of the simulation environment in [x, y, z]
    format are defined as:
  positive x points east,
```

```
1404    positive y points north,
1405    positive z points upward (sky).
1406
1407  Analyze the goal carefully and conceptualize a structure that can achieve
1408      this goal. Consider how the available blocks can be used to create
1409      this structure. Think about the physics and mechanics involved in
1410      achieving the goal.
1411
1412  Plan your structure by following these steps:
1413  1. Envision an overall structure that can achieve the goal.
1414  2. If necessary, break down this structure into non-redundant and
         reusable basic sub-structures or components, each sub-structure
         should be constructed independently, and the final structure will be
1415      assembled by attaching or connecting the sub-structures together.
1416  3. For each sub-structure, determine which building blocks will be used
1417      and how they will be arranged.
1418  4. Consider how these sub-structures will be assembled to form the
1419      complete structure.
1420  5. Think about how the complete structure will function to achieve the
         goal.
1421  6. Carefully compute the physical dimensions of the building blocks and
1422      the overall structure to ensure the structure is feasible without any
1423       overlap or conflict.
1424  7. The structures are mainly constructed by attaching a new block to the
1425      center of an un-occupied face of an existing block, so you should
1426      consider the relative position of the new block to the existing block
         .
1427  8. The attachment itself already has a connection with certain strength,
1428      brace is not necessary for the attachment, its only used to enhance
1429      the connection between two blocks that are already connected together
         , or to assemble structures that are not connected.
1430
1431  Your final output should be structured in the following format:
1432
1433  <building_plan>
1434  <overall_structure>
1435    <description>
        [Provide a detailed description of the overall structure]
1436    </description>
1437    <functionality>
1438      [Explain how this structure works to achieve the target goal]
1439    </functionality>
1440    <assembly>
1441      [Describe how the sub-structures are assembled to form the complete
            structure if multiple sub-structures are specified]
1442    </assembly>
1443    <motion_control>
1444      [Describe the motion control and the expected motion behavior of the
            structure to achieve the target goal if the structure is expected
1445       to move]
1446    </motion_control>
1447  </overall_structure>
1448
1449  <sub_structures>
1450    [For each sub-structure, include the following]
1451    <sub_structure_[number]>
1452      <name>[Name of the sub-structure]</name>
        <description>[Conceptual description of the sub-structure]</
1453         description>
1454      <components>[List of building blocks used]</components>
1455      <assembly>[How the components are arranged in the final structure if
            multiple sub-structures are specified]</assembly>
1456      <motion_control>[The expected motion control of the sub-structure to
1457         achieve the target goal if the sub-structure is expected to move
            ]</motion_control>
```

```
      <function>[The role this sub-structure plays in achieving the overall
          goal]</function>
      <design_requirements>[Overall design requirements for this sub-
          structure]</design_requirements>
  </sub_structure_[number]>
  [Repeat for each sub-structure]
</sub_structures>
</building_plan>

Remember, your final output should only include the content within the <
    building_plan> tags.
Ensure that your plan is detailed, logical, and clearly explains how the
    proposed structure will achieve the given goal using the available
    building blocks.

- Feasibility over optimality. Produce any workable plan; do not optimize
    part count or steps unless specified.
- Explicitly include in 'design_requirements': "Positions may be micro-
    adjusted in later stages to resolve conflicts based on actual build
    execution."
```

Drafter:

```
You are a Drafter who designs detailed blueprints of provided machine
    descriptions following these requirements:

  - The global coordinates in [x, y, z] format are defined as:
    positive x points east,
    positive y points north,
    positive z points upward (sky).

  - There will be a default 1x1x1 cubic starting block as the base at the
      beginning.
    There are {available_blks} blocks available. You must only use these
      blocks.

  - Provide a detailed illustration of the machine meeting the
      requirements. You MUST declare all blocks in your design, and you
      MUST follow the given format.

  - For **static blocks** (blocks without motion or non-structural
      functions), describe placement **relative to the previous block**
      using compass faces (e.g., north, south, top, bottom).
    Format:
    '<block i> - <block type> - <block note: a brief description of the
        block> - <relative position: which face (compass) of the previous
        block>'

  - For **functional blocks** (blocks with motion or structural functions
      ), provide extra information to describe the function and motion
      behavior of the block (e.g. a wheel that rolls towards the north, a
      cannon that shoots towards the south).
    Format:
    '<block i> - <block type> - <block note: a brief description of the
        block> - <relative position: which face (compass) of the previous
        block> - <function and motion behavior>'

  - The machine is constructed by placing each new block at the center of
      an unoccupied face of an existing block.

  - The coordinates of the blocks can be adjusted but mainly determined
      by the previous block.

  - You may argue with the reviewer for better solutions.
```

```
    - Your job is to translate the planner's plan into a buildable
       blueprint.
    - You may make position adjustments when the reviewer flags potential
       overlaps or when later build execution reveals conflicts.
    - Whenever you adjust, include a short **position adjustment note**
       describing what moved and why (e.g., "offset front axle +1 on X to
       clear chassis"), with flexibility **as needed per actual build
       execution**.
    - Do not change functional intent.
```

Reviewer:

```
You are a Reviewer who reviews blueprints of provided machine
    descriptions following these strict requirements:

STRUCTURAL REQUIREMENTS:

- The blueprint will be used to build the machine, so make sure the
    design is feasible and logical.
- There will be a default 1x1x1 shaped cubic starting block as the base
    at the beginning of the building process. Make sure the design has
    considered the base block.
- There are {available_blks}.
- For **each new block**, compute, check, and report:
1. The exact position (center coordinates) of the new block relative to
    the base block.
2. The distances between this new block's center and the centers of **all
    neighboring blocks** (blocks that have potential overlapping risks
    with the new block).
3. Whether any distance violates the minimum required distance (sum of
    half the block dimensions along the relevant axes).
- Any overlap or improper attachment must be flagged explicitly.

FUNCTIONAL VALIDATION:

- Check each point in detail, reasoning logically before proceeding to
    the next. Respond clearly whether the design meets or fails the
    requirement, and why.
1. Verify that the described structure allows the specified motion (e.g.,
    rotation, translation). State any missing or conflicting information
    that prevents confirmation.
2. For all functional components (e.g., wheels, cannon, etc.), carefully
    calculate their parameters (e.g., direction of motion, direction of
    shooting, etc.) and validate that they satisfy the functional
    requirements specified in the description (e.g., axis alignment,
    motion direction).
3. Verify moving components have appropriate mounting and alignment. Make
    sure their mounting and alignment are consistent with the expected
    motion behavior.

REVIEW PROCESS:

- First, **systematically check structural integrity and collision-free
    placement one block at a time** as outlined above.
- Then, validate functional implementation.
- Finally, assess physical feasibility.
- Only approve designs that pass all three checks.

Your review should present your analysis clearly in **step-by-step format
    **, showing your calculations and reasoning for each block.

If you believe the latest version of the blueprint has fully met the
    design requirements, please give your analysis to support this belief
    and include `TERMINATE` in your reply to finish the process.
```

```
- Prioritize feasibility over optimality. Check for overlaps, structural/
    functional conflicts, and ambiguous placement.
- When the design is acceptable, reply with `TERMINATE`. Otherwise, be
    specific about which placements likely collide or are under-
    constrained.
```

Builder:

```
You are an engineering building assistant server specialized in building
    functional structures in a simulated build environment following
    these requirements:

- You will be equipped with a series of tools to build the structure, and
     your role is to carefully follow the instruction from your
    collaborator, use suitable tools to fullfil the instruction, make
    suggestions of your tool during the conversation to help to
    accomplish the requirement of the collaborator.

- You MUST NOT make parallel tool calls, you can only make one tool call
    in your reply. Build the structure one block at a time.

- The simulation environment is a 3D space with a global coordinate
    system in [x, y, z] format, where positive x points east, positive y
    points north, and positive z points upward (sky).

- **IMPORTANT**: Start the conversation with a detailed introduction of
    all your tools, describe what they can accomplish, and what
    information you need to fully utilize them.

- Be sure to mention that the note argument of some tools is very
    important and useful to mark down the specific function of the block
    as a powerful identifier for the block.

- Execute guidance instructions step by step. Do not infer missing intent
     or change parts.
```

Guidance:

```
You are an engineering building engineer who gives step by step building
    instructions to build a functional structure in a simulated build
    environment:

- There are {available_blks}.
- The global coordinates of the simulation environment in [x, y, z]
    format are defined as:
    positive x points east,
    positive y points north,
    positive z points upward (sky).
- You will be provided with a design blueprint and a description of its
    functionality, your task is to determine the detailed building steps
    based on the blueprint.
- Make only one move in each reply to build the structure step by step,
    after the instruction is executed by the builder, you should analyze
    the latest structure feedback from the simulation environment, and
    decide the next step.
- If the execution of the instruction fails, you are encouraged to
    acquire the necessary information to analyze the failure, and give
    the next step instruction to correct the process.
- The building of the structure is mainly conducted by attaching a new
    block to an unoccupied face of an existing block, but you can also
    use other tools to adjust the structure if necessary.
- Ask the builder if you have any unclear information about the permitted
     building operations/tools.
```

```
- Do not be fully restricted to the blueprint, you can make some
    adjustment to the structure as long as it meets the design
    requirement and the structure can function as intended.
- There will be a default 1x1x1 shaped cubic starting block as the base
    at the beginning. This builder shall start the building process by
    initialize this block once the instruction is given.
- After you give the final step instruction, do not end the conversation
    yet, you MUST send the requirement to review the full structure at
    least once to make sure the final building process has been executed
    and the structure has been updated successfully.
- If you believe the latest structure is consistent to the blueprint,
    please give your analysis to support this belief and include '
    TERMINATE' in your reply to finish the process.

- You may make position adjustments during execution to resolve real
    collisions/constraints uncovered at build time, keeping functional
    intent intact.
- If repeated attempts still fail, use the available tool to **reject the
     current draft / request redesign**. Do this only after multiple good
    -faith tries.
- **IMPORTANT**: DO NOT make building related tools calls in your reply,
    your task is to give detailed step by step text instructions to the
    builder, the builder will execute the operations.
```

Controller:

```
You are a control engineer. Your job is to design control configurations
    and control sequences for a machine that will be tested in a
    simulation environment.
Your design must fulfill the given purpose while strictly following the
    task's evaluation protocol within 30 seconds.
The avaliable blocks in the simulation environment are:
<available_blocks>
{available_blks}
</available_blocks>

# Deliverable:

Return only one JSON object wrapped in a Markdown code block with the
    language tag json. Do not include any extra commentary before or
    after the code block.

```json
JSON_CONTENT
```

## control_design: string
A detailed analysis of the machine's structure and functionality
    according to the task's evaluation protocol. Explain how you will
    control the machine to fulfill the purpose, including assumptions,
    key constraints, and failure modes to avoid.

## control_config: list of objects
- Each object binds one key to one action on a specific block:

- key: string - must be one of:

  "UpArrow", "DownArrow", "LeftArrow", "RightArrow",

  "Alpha#" where # is 0-9 (e.g., "Alpha0", "Alpha7"),

  "Keypad#" where # is 0-9 (e.g., "Keypad3").

- action: string - the action you want this key to trigger, it MUST be
    one of the actions listed in the machine summary.
```

```
- block_id: string | integer - the identifier of the block the action
    applies to.

## control_sequence: list of objects
- Each object schedules a command on the timeline:

 - motion_action: string - a clear, detailed description of the commanded
     behavior, its purpose, and how it is implemented (should reference
     a key/action from control_config).

 - time: number - simulation time in seconds when the key is pressed.
     Must be >= 0. Use floating-point if needed.

 - key: string - must be a key defined in control_config.

 - hold_for: number - how long to hold the key in seconds. Must be > 0.

# Rules × Constraints

## Control configurations

- You bind keys to actions on powered blocks.

- The same key may control multiple actions simultaneously (across one or
     more blocks). For example, the key "Alpha1" may control the action "
     spinning_forward" of block 1 and the action "spinning_backward" of
     block 2 at the same time.

- A given action on a given block can also be controlled by multiple keys
     . For example, the action "spinning_forward" of block 1 can be
     controlled by the keys "Alpha1" and "Alpha2" at the same time.

## Control sequences

- Adding a sequence entry means: at time, press key, hold it for hold_for
     to activate the bound actions, then release it. For example, the
     sequence entry { "time": 1.0, "key": "Alpha1", "hold_for": 1.0 }
     means: at 1.0 seconds, press the key "Alpha1", hold it for 1.0
     seconds to activate the bound actions, then release it.

- Sequence entries may overlap in time.

- The machine will execute pressed keys by invoking all actions bound to
     those keys in control_config.

- Sort control_sequence by ascending time. Use consistent units (seconds)
     .

- The simulation only proceeds for 30 seconds, any actions beyond 30
     seconds will be ignored.

## Quality Bar

- Be task-driven: tie decisions explicitly to the evaluation protocol and
     the purpose.

- Be specific and measurable: include thresholds, margins, and safety
     checks when relevant.

- State assumptions if required inputs are missing, but keep them
     realistic and minimal.

- Prefer concise technical language; avoid fluff.
```

```
- Ensure internal consistency between control_config and control_sequence
    (keys used in sequences must exist in the config; actions referenced
    must be bound as specified).

- Output must contain valid JSON and wrapped exactly as the following
    format:

```json
{ "control_design": "str, The detailed analysis of the machine's
    structure and functionality according to the evaluation protocol of
    the task, explain the how you would like to control the machine to
    fulfill the given purpose",
"control_config":
  [
    {
      "key": "str, The key you decide to use, it must be one of the keys
          ['UpArrow', 'DownArrow', 'LeftArrow', 'RightArrow', 'Alpha#', '
          Keypad#'] where # is a number from 0 to 9",
      "action": "str, The action you want to use for the key",
      "block_id": "str | int, The block id of which the action is applied
          "
    }
  ],
"control_sequence":
  [
    {
      "motion_action": "str, The detailed explanation of the action you
          want take, what is the purpose of this action and how to
          implement it",
      "time": "float, The time you decide to press the key",
      "key": "str, The key you decide to press, it must be one of the
          keys in your control config",
      "hold_for": "float, The duration you decide to press the key in
          seconds"
    }
  ]
}
```
# Control, Simulation, and Revision

- The control_config and control_sequence are the control configurations
    and sequences that guide the machine's actions.
- The simulation is the simulated motion trajectory of the machine,
    describing the motion trajectory (x, y, z) of some blocks in the
    machine.
- You should analyze the simulation and the control_config and
    control_sequence to revise the control_config and control_sequence to
    optimize the task.
```

## I MORE EXPERIMENT RESULTS

**Multi-Agent Pipeline Ablation:** To validate the necessity of our five-role workflow design, we compare it against simpler controller variants on the **Support** Lv.1 task using the Seed-1.6 model. As shown in Table 4, the full five-role workflow (Planner, Drafter, Reviewer, Guidance, Builder) substantially outperforms simplified alternatives. The Guidance-Builder variant, which removes the planning and review stages, achieves only a 4.7% success rate ($-40.6\%$) with significantly lower maximum load capacity. The single-agent Builder baseline performs better than Guidance-Builder in success rate (22.6%) but suffers from an extremely high invalid action rate of 44.7% ($+43.3\%$), indicating severe difficulty in generating valid construction sequences without structured guidance. These results demonstrate that the layered five-role architecture effectively mirrors the natural task structure: global planning establishes machine type, draft-review cycles filter design flaws, guidance

decomposes blueprints into executable steps, and the builder performs actions, together maintaining low invalid rates (1.4%) while achieving substantially higher construction success and performance.

Table 4: Ablation study on multi-agent pipeline for Seed-1.6 model on **Support** Lv.1 task ($n = 64$ samples). The Indicator measures the maximum load the bridge can support. Metrics are averaged across samples. Performance changes relative to our workflow are shown in parentheses.

| Workflow | Number of Parts | Success Rate (%)↑ | Indicator (Max Load)↑ | Invalid-Action Rate (%)↓ |
|---|---|---|---|---|
| **Five-role Multi-agent** | **33.4** | **45.3** | **197.4** | **1.4** |
| Guidance-Builder | 20.2 | 4.7 ($-40.6$) | 31.4 ($-166.0$) | 1.7 ($+0.3$) |
| Builder | 19.8 | 22.6 ($-22.7$) | 75.5 ($-121.9$) | 44.7 ($+43.3$) |

**Decoding Sensitivity:** To assess the robustness of our pipeline to decoding configurations, we conduct ablation experiments varying temperature settings on the **Support** Lv.1 task using the Seed-1.6 model. As shown in Table 5, our default configuration ($temperature = 0.5$, top-p $= 0.7$) achieves the highest success rate of 45.3%. Varying temperature shows modest impact: $temperature = 0.0$ yields 43.8% success rate with slightly higher invalid actions (2.3%), while $temperature = 1.0$ achieves the highest maximum load indicator (199.1) but lower success rate (42.2%). Overall, performance variations across configurations remain within a reasonable range, with invalid action rates consistently low, demonstrating that the benchmark remains discriminative and solvable across different decoding settings.

Table 5: Ablation study on decoding strategies for Seed-1.6 model on **Support** Lv.1 task ($n = 64$ samples). Bold indicates our default configuration. Underline indicates the best performance for each metric.

| Temperature | Top-p | Number of Parts | Success Rate (%)↑ | Indicator (Max Load)↑ | Invalid-Action Rate (%)↓ |
|---|---|---|---|---|---|
| **0.5** | **0.7** | **33.4** | **45.3** | **197.4** | **1.4** |
| 0.0 | 0.7 | 30.0 | 43.8 | 168.8 | 2.3 |
| 1.0 | 0.7 | 29.8 | 42.2 | 199.1 | 1.8 |

**Closed-loop Feedback Refinement:** To evaluate the potential of iterative refinement through closed-loop feedback, we conducted additional experiments on the **Support** Lv.1 task. We randomly selected 18 failed samples from the Seed-1.6 model and incorporated simulation results as feedback for subsequent refinement turns. As shown in Table 6, closed-loop feedback substantially improves construction performance: 72% (13/18) of samples pass after the first refinement turn, increasing to 83% (15/18) after Turn 3, and reaching a 100% (18/18) success rate by Turn 5. It demonstrates that integrating simulation feedback enables models to iteratively correct design flaws and meet task requirements. However, the performance gain comes at the cost of increased computational expense due to multiplied LLM inference rounds. While closed-loop refinement is valuable for practical engineering applications, we adopt single-turn evaluation as the default benchmark setting to better distinguish the inherent construction capabilities across different LLMs.

Table 6: Closed-loop feedback refinement results on **Support** Lv.1 task ($n = 18$ failed samples). The Indicator measures the maximum load the bridge can support. Metrics are averaged across samples at each refinement turn.

| Metric | Round 1 | Round 2 | Round 3 | Round 4 | Round 5 |
|---|---|---|---|---|---|
| Success Rate (%)↑ | 0.0 | 72.2 | 83.3 | 94.4 | 100 |
| Indicator (Max Load)↑ | 0.0 | 422.6 | 477.2 | 568.8 | 586.9 |
| Number of Parts | 26.5 | 61.0 | 49.8 | 82.7 | 43.0 |
| Invalid-Action Rate (%)↓ | 1.2 | 1.1 | 2.2 | 0.9 | 1.2 |
| # LLM Requests↓ | 56.8 | 128.8 | 108.2 | 171.7 | 89.0 |

## J    ANALYSIS OF SPATIAL CONFLICTS

Current LLMs acquire capabilities mainly through two stages: pre-training and post-training. They memorize knowledge during pre-training and develop reasoning skills in post-training. Our analysis identifies that existing LLMs lack spatial capabilities, as evidenced by frequent spatial conflicts. However, a critical constraint hinders in-depth analysis of the underlying causes as key information about frontier models, such as their data, architecture, and training processes, is generally unavailable. We therefore propose the following two hypotheses corresponding to the two stages, respectively:

- Poor pre-training effectiveness. Engineering construction involves complex spatial imagination and reasoning. When humans learn related knowledge, they usually rely on schematics of the construction process. Learning solely through textual descriptions is difficult even for humans. However, LLMs are trained on pure text, leading to inadequate training results.

- In the post-training stage, existing LLMs mainly focus on reasoning tasks like mathematics and coding. They may lack appropriate environments, verifiable rewards, and efficient learning methods to develop long-range 3D spatial reasoning and construction capabilities. Even if some models have strong 3D spatial reasoning, construction typically requires dozens or even hundreds of reasoning steps. A single mistake in any step will lead to overall failure, resulting in low success rates.

Nevertheless, although we can make such hypotheses, it is impractical to verify them for closed-source LLMs.