# OpenReview forum: "BuildArena: A Physics‑Aligned Interactive Benchmark of LLMs for Engineering Construction"
_ICLR.cc/2026/Conference — ICLR 2026 Conference Desk Rejected Submission_

### Official Review · Reviewer_TqAt · 2025-10-22

**Soundness:** 3
**Presentation:** 2
**Contribution:** 1
**Rating:** 2
**Confidence:** 3

**Summary:**

This paper introduces BuildArena, a physics-aligned benchmark designed to evaluate LLMs on engineering construction tasks. The framework provides three task categories (Support, Transport, Lift) of varying difficulty, an open-source library to interact with the Besiege physics simulator via language, and a baseline multi-agent workflow. Experiments on eight frontier LLMs demonstrate that while models show basic competence, they systematically fail at tasks requiring precision, hierarchical assembly, and robust spatial reasoning.

**Strengths:**

1: The work addresses a gap by proposing the first benchmark for evaluating LLMs on physically-grounded, multi-step assembly. The problem is well-motivated.

2: The tasks are systematically designed around clear engineering difficulty dimensions (e.g., precision, compositionality). The evaluation is thorough, with extensive trials and an insightful breakdown of failure modes, which provides clear takeaways.

3: The 3D Spatial Geometric Computation Library is a nice engineering contribution that enables this line of research and allows the community to build upon it.

**Weaknesses:**

1: The paper is best understood as a benchmarking paper. Its main contribution is the evaluation framework itself, not a new AI methodology. The multi-agent workflow is an application of existing patterns (e.g., AutoGen-style collaboration) and feels more like a necessary piece of engineering to enable the experiments rather than a novel contribution.

2: The analysis shows that models fail and what they fail on (e.g., spatial conflict), but it lacks a deep dive into why. For instance, it doesn't offer hypotheses or analysis on what specific training data, architectural choices, or fine-tuning techniques might explain the performance gap between models. The paper stops short of suggesting how to improve LLMs for this domain.

3: The benchmark only tests one-shot construction. A key engineering skill is iterative refinement based on testing and failure analysis. By not including a "closed-loop" where the model receives simulation feedback to fix its design.

**Questions:**

The analysis shows Grok-4 is exceptionally good at precision tasks. Do authors have any hypotheses about its architecture or training that might explain this significant advantage over other models?

The primary failure mode is "spatial conflict." Do authors see this as a fundamental limitation of current LLMs in tracking world state, or a problem that could be addressed with better prompting or specialized tool use?

---

> ### Author Response · Authors · 2025-11-23
>
> We thank the reviewer for the helpful comments. We are glad that the reviewer acknowledges our work is well-motivated. The reviewer also appreciates the systematically designed tasks and thorough, extensive, and insightful evaluations. Additionally, the reviewer highlights the notable engineering contribution of our 3D Spatial Geometric Computation Library, which facilitates future community research. Below, we address the reviewer's concerns one by one.
>
> > Comment 1: The paper is best understood as a benchmarking paper. Its main contribution is the evaluation framework itself, not a new AI methodology. The multi-agent workflow is an application of existing patterns (e.g., AutoGen-style collaboration) and feels more like a necessary piece of engineering to enable the experiments rather than a novel contribution.
> >
>
> **Reply**: Thanks for pointing out this confusion. Our main contribution is that we propose the first physics-aligned interactive benchmark designed to assess LLMs' capabilities in engineering construction tasks. To achieve this goal, we develop a multi-agent workflow, which is referred to as "technical contribution" in our original submission, meaning that it is a supporting element rather than a core contribution. We have revised the last paragraph of the Introduction Section of the manuscript accordingly to remove this confusion.
>
> > Comment 2: The analysis shows that models fail and what they fail on (e.g., spatial conflict), but it lacks a deep dive into why. For instance, it doesn't offer hypotheses or analysis on what specific training data, architectural choices, or fine-tuning techniques might explain the performance gap between models. The paper stops short of suggesting how to improve LLMs for this domain.
> >
>
> **Reply**: Current LLMs acquire capabilities mainly through two stages: pre-training and post-training. They memorize knowledge during pre-training and develop reasoning skills in post-training. Our analysis identifies that existing LLMs lack spatial capabilities, as evidenced by frequent spatial conflicts. However, a critical constraint hinders in-depth analysis of the underlying causes as key information about frontier models, such as their data, architecture, and training processes, is generally unavailable. We therefore propose the following two hypotheses corresponding to the two stages, respectively:
>
> 1. Poor pre-training effectiveness. Engineering construction involves complex spatial imagination and reasoning. When humans learn related knowledge, they usually rely on schematics of the construction process. Learning solely through textual descriptions is difficult even for humans. However, LLMs are trained on pure text, leading to inadequate training results.
> 2. In the post-training stage, existing LLMs mainly focus on reasoning tasks like mathematics and coding. They may lack appropriate environments, verifiable rewards, and efficient learning methods to develop long-range 3D spatial reasoning and construction capabilities. Even if some models have strong 3D spatial reasoning, construction typically requires dozens or even hundreds of reasoning steps. A single mistake in any step will lead to overall failure, resulting in low success rates.
>
> Nevertheless, although we can make such hypotheses, verifying them is impractical for closed-source LLMs. We have further incorporated the discussion into the revised manuscript's Appendix J.
>
> > Comment 3: The analysis shows Grok-4 is exceptionally good at precision tasks. Do authors have any hypotheses about its architecture or training that might explain this significant advantage over other models?
> >
>
> **Reply**: Grok 4 has no public official technical report. Only a few technical details have been disclosed on its official website, based on which its strong performance may be attributed to the following three key innovations:
>
> 1. Native tool use. This feature may allow the model to call external mathematical verification tools for tasks requiring rigorous geometric calculations, such as spatial reasoning in design.
> 2. Integrating reinforcement learning into the pre-training stage, with its scope expanded from mathematics and coding to more fields. This gives the model stronger reasoning capabilities.
> 3. Powerful computing resources. It uses a cluster of 200,000 GPUs, with training leveraging over an order of magnitude more computing power than previous models.

---

> > ### Author Response · Authors · 2025-11-23
> >
> > > Comment 4: The benchmark only tests one-shot construction. A key engineering skill is iterative refinement based on testing and failure analysis. By not including a "closed-loop" where the model receives simulation feedback to fix its design.
> > >
> >
> > **Reply**: We agree that closed-loop feedback plays an important role in engineering construction and optimization. To test the performance gain by integrating the closed-loop feedback, we conduct additional experiments. We randomly select 18 samples from the failure cases of Seed-1.6 model in the Support-Lv.1 task, and add the latest machine structure and simulation results as feedback to the workflow's initial input of next turn, achieving closed-loop engineering construction. As shown in the following Table 7, closed-loop feedback significantly improves the performance of construction outcomes, with most samples meeting test standards after the first refinement round and all passing after four more rounds. However, this improvement comes at the cost of increased inference expenses (token count).
> >
> > Table 7: Multi-turn closed-loop construction results of Seed-1.6 model on Support-Lv.1 task (Note: Round 1 means no closed-loop).
> >
> > |  | Round 1 | Round 2 | Round 3 | Round 4 | Round 5 |
> > | --- | --- | --- | --- | --- | --- |
> > | Success Rate (%) ↑ | 0.0 | 72.2 | 83.3 | 94.4 | 100 |
> > | Indicator (Max Load) ↑ | 0.0 | 422.6 | 477.2 | 568.8 | 586.9 |
> > | Number of Parts | 26.5 | 61.0 | 49.8 | 82.7 | 43.0 |
> > | Invalid-Action Rate (%) ↓ | 1.2 | 1.1 | 2.2 | 0.9 | 1.2 |
> > | # LLM Requests ↓ | 56.8 | 128.8 | 108.2 | 171.7 | 89.0 |
> >
> > Meanwhile, we are conducting ongoing experiments on Support-Lv.3 task. Sampling on this task takes much more time due to the increased task difficulty. We promise to update the results on this page within one week, and these results and corresponding analysis will also be added to the manuscript.
> >
> > Current experimental results in Table 7 above show that closed-loop sampling enhances models' construction capabilities as the number of rounds increases. However, this is a benchmark paper aiming to distinguish the capabilities of different LLMs clearly. If multi-round sampling is adopted for all models, their final performance may become less distinguishable. Therefore, we adopt single-round evaluation as the default setting in our paper.
> >
> > > Comment 5: The primary failure mode is "spatial conflict." Do authors see this as a fundamental limitation of current LLMs in tracking world state, or a problem that could be addressed with better prompting or specialized tool use?
> > >
> >
> > **Reply**:
> >
> > It is well known that LLMs exhibit hallucinations, and due to the lack of 3D spatial data in the training process, this issue is particularly severe for 3D spatial reasoning [7,8,9]. The generation process itself does not fundamentally forbid content that describes structures violating physical constraints. For example, in a vehicle-construction task, LLMs may sample a plan that places two wheels of radius 1 at an axle distance of 1 (where the minimum distance should be 2), which leads to spatial intersection of the wheels. Our developed Spatial Geometric Computation Library is used to numerically check and filter out such invalid actions, ensuring that the final constructed structure obeys the physical constraints. However, this only corrects errors post hoc and does not prevent the LLM from sampling physically inconsistent text in the first place. This reflects a fundamental limitation of current LLMs in physics-constrained 3D reasoning.
> >
> > Reference
> >
> > [7] Zha et al. How to Enable LLM with 3D Capacity? A Survey of Spatial Reasoning in LLM. arXiv preprint 2025.
> >
> > [8] Rodionov et al. PlanQA: A Benchmark for Spatial Reasoning in LLMs using Structured Representations. arXiv preprint 2025.
> >
> > [9] Feng et al. A Survey of Large Language Model-Powered Spatial Intelligence Across Scales: Advances in Embodied Agents, Smart Cities, and Earth Science. arXiv preprint 2025

---

> > > ### Comment · Reviewer_TqAt · 2025-11-25
> > >
> > > Thanks to the authors for preparing the response with some insightful points. I have updated my score based on the changes in the response.

---

> ### Author Response · Authors · 2025-12-03
>
> Following our previous update, we have completed the additional experiments on the high-difficulty Support-lv3 task. The results (Table 11) present a contrasting and intriguing perspective compared to the simpler tasks. Unlike Support-lv.1, where closed-loop feedback secured a 100% success rate, the Support-lv.3 samples remained at **0/18 success across all 5 turns** of refinement.
>
> Table 11: Multi-turn closed-loop construction results of Seed-1.6 model on Support-Lv.3 task (Note: Round 1 means no closed-loop).
>
> |  | Round 1 | Round 2 | Round 3 | Round 4 | Round 5 |
> | --- | --- | --- | --- | --- | --- |
> | Success Rate (%) ↑ | 0.0 | 0.0 | 0.0 | 0.0 | 0.0 |
> | Indicator (Max Load) ↑ | 0.0 | 0.0 | 0.0 | 0.0 | 0.0 |
> | Number of Parts | 98.0 | 52.6 | 66.2 | 69.9 | 58.5 |
> | Invalid-Action Rate (%) ↓ | 16.4 | 17.5 | 16.5 | 12.2 | 14.7 |
> | # LLM Requests ↓ |197.1 | 224.7 | 216.3 | 183.8 | 283.9 |
>
> We observe that for highly complex structural problems, early failure modes often narrow the decision space. The model tends to "patch" a fundamentally flawed design rather than reimagine it in a whole new perspective, effectively getting trapped in local optima. While these 18 refined seeds failed, successful designs **do exist** within our dataset when using independent random sampling. For difficult tasks, **"starting over" (broad exploration) often shines where "fixing" (deep exploitation) tends to fail.**
>
> This finding aligns with recent literature on test-time compute, such as scaling inference-time searching [10], which demonstrates that scaling simple random sampling can be a highly effective strategy.
>
> Conclusion: Closed-loop refinement is not a universal guarantee of success; its efficacy is highly sensitive to task difficulty and the quality of the initial seed. Since relying on closed-loop could mask the model's ability to generate high-quality initial candidates, and does not consistently solve hard tasks. We reaffirm that **single-round evaluation** remains the most objective and consistent metric for distinguishing inherent model capabilities in this benchmark.
>
> [10] Zhao et al. Sample, Scrutinize and Scale: Effective Inference-Time Search by Scaling Verification. arXiv preprint 2025.

---

### Official Review · Reviewer_fcZf · 2025-10-30

**Soundness:** 3
**Presentation:** 3
**Contribution:** 2
**Rating:** 4
**Confidence:** 3

**Summary:**

The paper proposes BuildArena, a physics-aligned, interactive benchmark to test whether LLMs can transform natural-language instructions into physically feasible 3D constructions. It introduces (i) a customizable framework with three task families—Support, Transport, Lift—each with three difficulty levels, (ii) an open 3D Spatial Geometric Computation Library mirroring Besiege’s build logic, and (iii) a five-role agentic workflow for plan-draft-review-build guidance. Experiments on eight frontier LLMs show non-trivial but limited abilities (e.g., low success on high-precision Lift), with Grok-4 strongest overall.

**Strengths:**

The work fills a clear gap by jointly evaluating language → spatial reasoning → physics feasibility rather than isolated planning or static reasoning. The benchmark is thoughtfully designed (multi-task, multi-level, logged simulations) and reasonably clear, with concrete metrics per task and illustrative figures for the pipeline and workflow; the released code further supports adoption. Empirically, the analysis (tables/figures on success, failure modes, and cost) surfaces stable weaknesses in compositionality/precision and suggests that more tokens do not automatically yield better constructions.

**Weaknesses:**

- Scope realism. Besiege-style modular assembly is still “toy engineering”; claims about engineering automation should be scoped more conservatively.
- Motivation→evidence gap. The paper claims to “comprehensively evaluate diverse capabilities,” but provides no ablations over prompts, outer loops, or alternative workflows to show the pipeline itself is necessary and beneficial.
- Baselines/ablations. No comparisons to simpler agents (e.g., single-LLM ReAct/Plan-Execute) or workflow ablations (remove Reviewer/Guidance, fewer turns).
- External validity. It is unclear how results transfer to CAD/robotic assembly engines (e.g., MuJoCo/Isaac Gym) or to non-modular component sets beyond Besiege.

**Questions:**

- **Primary contribution**. Is the central contribution the benchmark, the open geometric library, the five-role LLM workflow, or the end-to-end pipeline? Please state the primary claim explicitly and rank the others as supporting elements.
- **Pipeline necessity**. Does the five-role workflow significantly outperform simpler controllers (single-agent ReAct / Plan-Execute)? Provide ablations removing Reviewer or Guidance and limiting loop depth, with ∆success/indicator/invalid-action rates.
- **Prompt/decoding sensitivity**. How sensitive are outcomes to prompt variants, few-shot vs. zero-shot, and decoding (temperature, top-p)? Please report variance to establish pipeline robustness rather than prompt idiosyncrasy.
- **Transfer**: Any evidence BuildArena policies/skills transfer to CAD or to robot-assembly environments (MuJoCo/Isaac Gym)?

---

> ### Author Response · Authors · 2025-11-23
>
> We thank the reviewer for the valuable and detailed comments. We are grateful that the reviewer recognizes our work addresses a clear research gap. The reviewer also appreciates the benchmark for its thoughtful design, clear logic and ease of adoption with the released code. Below, we address the reviewer's concerns one by one.
>
> > Comment 1: Scope realism. Besiege-style modular assembly is still "toy engineering"; claims about engineering automation should be scoped more conservatively.
> >
>
> **Reply**: We agree that Besiege represents a simplified, modular form of engineering and does not aim to capture the full realism of engineering automation. Thus, we are glad to take your suggestion and conservatively claim our scope as "a first step towards engineering automation using LLMs". We have modified the Abstract and Introduction section in the revised manuscript accordingly.
>
> > Comment 2: Motivation→evidence gap. The paper claims to "comprehensively evaluate diverse capabilities," but provides no ablations over prompts, outer loops, or alternative workflows to show the pipeline itself is necessary and beneficial.
> >
>
> **Reply**: We agree that ablation studies on these factors are important and we have conducted additional experiments on each of them.
>
> For outer loops, as we understand, it means using closed-loop feedback from the simulator to improve the construction performance iteratively. To test the performance gain by integrating such closed-loop feedback, we randomly select 18 samples from the failure cases of Seed-1.6 model the Support-Lv.1 task, and add the latest machine structure and simulation results as feedback to the workflow's initial input of next turn, achieving closed-loop engineering construction. As shown in the following Table 4, closed-loop feedback significantly improves the performance of construction outcomes, with most samples meeting test standards after the first refinement round and all passing after four more rounds. However, this improvement comes at the cost of increased inference expenses (token count).
>
> Table 4:  Multi-turn closed-loop construction results of Seed-1.6 model on Support-Lv.1 task (Note: Round 1 means no closed-loop).
>
> |  | Round 1 | Round 2 | Round 3 | Round 4 | Round 5 |
> | --- | --- | --- | --- | --- | --- |
> | Success Rate (%) ↑ | 0.0 | 72.2 | 83.3 | 94.4 | 100 |
> | Indicator (Max Load) ↑ | 0.0 | 422.6 | 477.2 | 568.8 | 586.9 |
> | Number of Parts | 26.5 | 61.0 | 49.8 | 82.7 | 43.0 |
> | Invalid-Action Rate (%) ↓ | 1.2 | 1.1 | 2.2 | 0.9 | 1.2 |
> | # LLM Requests ↓ | 56.8 | 128.8 | 108.2 | 171.7 | 89.0 |
>
> Meanwhile, we are conducting ongoing experiments on Support-Lv.3 task. Sampling on this task takes much more time due to the increased task difficulty. We promise to update the results on this page within one week, and these results and corresponding analysis will also be added to the manuscript.
>
> Current experimental results in Table 4 above show that closed-loop sampling enhances models' construction capabilities as the number of rounds increases. However, this is a benchmark paper aiming to clearly distinguish the capabilities of different LLMs. If multi-round sampling is adopted for all models, their final performance may become less distinguishable. Therefore, we adopt single-round evaluation as the default setting in our paper.
>
> For ablations over alternative workflows, please refer to our reply to the next Comment 3.
>
> For consideration of prompts, please refer to our reply to the next Comment 6.

---

> ### Author Response · Authors · 2025-11-23
>
> > Comment 3: Baselines/ablations. No comparisons to simpler agents (e.g., single-LLM ReAct/Plan-Execute) or workflow ablations (remove Reviewer/Guidance, fewer turns). Does the five-role workflow significantly outperform simpler controllers (single-agent ReAct / Plan-Execute)? Provide ablations removing Reviewer or Guidance and limiting loop depth, with ∆success/indicator/invalid-action rates.
> >
>
> **Reply**: Yes, we tested simpler controllers, including a single-agent ReAct-style setup and a two-stage Plan-Execute controller. The results in Table 5 show that these variants perform worse than the 5-Agent baseline. In particular, the single-agent setup produces extremely high invalid-action rates.
>
> The five-role workflow is more stable because it mirrors the natural structure of the task. The planner proposes a global machine type, the draft–reviewer pair checks the blueprint and filters out issues, the guidance agent turns this into step-wise actions, and the builder executes them. This layered process avoids error cascades and keeps invalid actions consistently low across tasks. Therefore, it provides a practical and reliable default controller for the benchmark. Our contribution is to provide a discriminative benchmark rather than mandate a specific pipeline design.
>
> Table 5: Ablation study on multi-agent pipeline for Seed-1.6 model on Support-Lv.1 task. Performance changes relative to the 5-Agent baseline are shown in parentheses.
>
> | Workflow | Number of Parts | Success Rate (%) ↑ | Indicator (Max Load) ↑ | Invalid-Action Rate (%) ↓ |
> | --- | --- | --- | --- | --- |
> | **5-Role-Multi-Agent** | **33.4** | **45.3** | **197.4** | **1.4** |
> | Guidance-Builder | 20.2 | 4.7(-40.6) | 31.4 (-166.0) | 1.7(+0.3) |
> | Builder | 19.8 | 22.6 (-22.7) | 75.5 (-121.9) | 44.7(+43.3) |
>
> > Comment 4: External validity. It is unclear how results transfer to CAD/robotic assembly engines (e.g., MuJoCo/Isaac Gym) or to non-modular component sets beyond Besiege. Any evidence of BuildArena policies/skills transfer to CAD or to robot-assembly environments (MuJoCo/Isaac Gym)?
> >
>
> **Reply**: We did not use other high-fidelity simulators like MuJoCo. Our paper aims to build a benchmark for testing LLM's construction capabilities, thus the simulator must meet three necessary conditions: (1) support step-by-step construction, (2) the construction process complies with physical laws, (3) the functions related to construction are open-source or can be encapsulated as API calls. Our adopted Besiege is the only one that meets these conditions. Although MuJoCo is a high-fidelity physical simulator, it is primarily used to simulate the dynamic behaviors of already constructed mechanical systems, rather than supporting interactive construction processes. Therefore, it is not suitable for our task.
>
> We believe BuildArena policies/skills could be transferred to CAD or robot-assembly environments. This is because during the construction process, LLMs only interact with the text descriptions of the environment's parameterized module library and geometric computation functions. As long as an environment includes a sufficiently diverse set of modules and supports a rich range of action functions, the construction capabilities of LLMs demonstrated in BuildArena can be transferred to that environment.
>
> > Comment 5: Primary contribution. Is the central contribution the benchmark, the open geometric library, the five-role LLM workflow, or the end-to-end pipeline? Please state the primary claim explicitly and rank the others as supporting elements.
> >
>
> **Reply**: Our main contribution is described by only one sentence: **We propose the first physics-aligned interactive benchmark designed to assess LLMs' capabilities in engineering construction tasks**. To achieve this contribution, we develop two technical contributions: (1) an extendable task design strategy; (2) a 3D Spatial Geometric Computation Library. They are referred to as "technical contributions", meaning that they are supporting elements rather than core contributions. In our original submission, the end-to-end workflow was also referred to as a "technical contribution", now it is removed since it is not an essential component. We have revised the Abstract and the last paragraph of the Introduction Section of the manuscript accordingly to clarify such confusion.

---

> ### Author Response · Authors · 2025-11-23
>
> > Comment 6: Prompt/decoding sensitivity. How sensitive are outcomes to prompt variants, few-shot vs. zero-shot, and decoding (temperature, top-p)? Please report variance to establish pipeline robustness rather than prompt idiosyncrasy.
> >
>
> **Reply**: As described in our original submission, our task prompts follow a strictly zero-shot design and do not include any task-level few-shot examples. For details about task prompts, please refer to Appendix G of our original submission. Manually crafted few-shot examples would constrain the diversity of generated constructions and limit performance [6]. Such settings prevent the benchmark from reliably evaluating the LLMs' design and construction capabilities. For this reason, we do not conduct few-shot experiments.
>
> Regarding the decoding sensitivity, we conduct experiments with altered decoding parameters. As shown in Table 6, our pipeline demonstrates strong stability across different decoding settings: varying these parameters has a minimal influence on success rates and key indicators.
>
> Table 6: Ablation study on decoding strategies for Seed-1.6 model on Support-Lv.1 task. Bold indicates our default configuration, copied from Table 2 in the original submission.
>
> | Temperature (T) | Top-p | Number of Parts | Success Rate (%) ↑ | Indicator (Max Load) ↑ | Invalid-Action Rate (%) ↓ |
> | --- | --- | --- | --- | --- | --- |
> | **0.5** | **0.7** | **33.4** | **45.3** | **197.4** | **1.4** |
> | 0.0 | 0.7 | 30.0 | 43.8 | 168.8 | 2.3 |
> | 1.0 | 0.7 | 29.8 | 42.2 | 199.1 | 1.8 |
>
> Reference
>
> [6] Zhao et al. Calibrate Before Use: Improving Few-Shot Performance of Language Models. PMLR 2021.

---

> ### Author Response · Authors · 2025-12-03
>
> Following our previous update, we have completed the additional experiments on the high-difficulty Support-lv3 task. The results (Table 10) present a contrasting and intriguing perspective compared to the simpler tasks. Unlike Support-lv.1, where closed-loop feedback secured a 100% success rate, the Support-lv.3 samples remained at **0/18 success across all 5 turns** of refinement.
>
> Table 10: Multi-turn closed-loop construction results of Seed-1.6 model on Support-Lv.3 task (Note: Round 1 means no closed-loop).
>
> |  | Round 1 | Round 2 | Round 3 | Round 4 | Round 5 |
> | --- | --- | --- | --- | --- | --- |
> | Success Rate (%) ↑ | 0.0 | 0.0 | 0.0 | 0.0 | 0.0 |
> | Indicator (Max Load) ↑ | 0.0 | 0.0 | 0.0 | 0.0 | 0.0 |
> | Number of Parts | 98.0 | 52.6 | 66.2 | 69.9 | 58.5 |
> | Invalid-Action Rate (%) ↓ | 16.4 | 17.5 | 16.5 | 12.2 | 14.7 |
> | # LLM Requests ↓ |197.1 | 224.7 | 216.3 | 183.8 | 283.9 |
>
> We observe that for highly complex structural problems, early failure modes often narrow the decision space. The model tends to "patch" a fundamentally flawed design rather than reimagine it in a whole new perspective, effectively getting trapped in local optima. While these 18 refined seeds failed, successful designs **do exist** within our dataset when using independent random sampling. For difficult tasks, **"starting over" (broad exploration) often shines where "fixing" (deep exploitation) tends to fail.**
>
> This finding aligns with recent literature on test-time compute, such as scaling inference-time searching [10], which demonstrates that scaling simple random sampling can be a highly effective strategy.
>
> Conclusion: Closed-loop refinement is not a universal guarantee of success; its efficacy is highly sensitive to task difficulty and the quality of the initial seed. Since relying on closed-loop could mask the model's ability to generate high-quality initial candidates, and does not consistently solve hard tasks. We reaffirm that **single-round evaluation** remains the most objective and consistent metric for distinguishing inherent model capabilities in this benchmark.
>
> [10] Zhao et al. Sample, Scrutinize and Scale: Effective Inference-Time Search by Scaling Verification. arXiv preprint 2025.

---

### Official Review · Reviewer_mYoq · 2025-11-01

**Soundness:** 3
**Presentation:** 3
**Contribution:** 3
**Rating:** 6
**Confidence:** 4

**Summary:**

The authors introduce BuildArena as a first physics-aligned interactive benchmark for language-driven engineering construction. They construct three task categories - Support, Transport, and Lift. Furthermore, an open source version of spatial geometric computation library is presented. The benchmark is evaluated on eight closed source models.

**Strengths:**

- the proposed framework is comprehensive
- addresses an underexplored area (construction)
- development of an open source spatial geometric computation library is good for the research community

**Weaknesses:**

- the analysis remains mostly in an aggregate form and qualitative breakdown is missing. In fig 6, it would be good to have insights into why spatial conflicts occur.
- as mentioned in the limitations section, the framework does not have a feedback loop between the simulator results and construction results.
- the evaluation seems to conflate model capabilities with the specific 5-agent workflow and I do not know if the poor performance reflects model limitations or suboptimal orchestration.

**Questions:**

1. How do you ensure physics consistency between your spatial geometric library and Beseige’s internal engine?
2. Have you tried using a higher fidelity engine (maybe MuJoCo)?
3. Have you tested simpler workflows? How sensitive are the results to the 5-agent setup you currently adopt?

---

> ### Author Response · Authors · 2025-11-23
>
> We thank the reviewer for the valuable and detailed comments. We are grateful that the reviewer recognizes our work addresses an underexplored area (construction), with a comprehensive framework and an open-source spatial geometric computation library that is good for the research community. Below, we address the reviewer's questions one by one.
>
> > Comment 1: The analysis remains mostly in an aggregate form and qualitative breakdown is missing. In fig 6, it would be good to have insights into why spatial conflicts occur.
> >
>
> **Reply**:
>
> It is well known that LLMs exhibit hallucinations, and due to the lack of 3D spatial data in the training process, this issue is particularly severe for 3D spatial reasoning [3,4,5]. The generation process itself does not fundamentally forbid content that describes structures violating physical constraints. For example, in a vehicle-construction task, LLMs may sample a plan that places two wheels of radius 1 at an axle distance of 1 (where the minimum distance should be 2), which leads to spatial intersection of the wheels. Our developed Spatial Geometric Computation Library is used to numerically check and filter out such invalid actions, ensuring that the final constructed structure obeys the physical constraints. However, this only corrects errors *post hoc* and does not prevent the LLM from sampling physically inconsistent text in the first place. This reflects a fundamental limitation of current LLMs in physics-constrained 3D reasoning.
>
> Reference
>
> [3] Zha et al. How to Enable LLM with 3D Capacity? A Survey of Spatial Reasoning in LLM. arXiv preprint 2025.
>
> [4] Rodionov et al. PlanQA: A Benchmark for Spatial Reasoning in LLMs using Structured Representations. arXiv preprint 2025.
>
> [5] Feng et al. A Survey of Large Language Model-Powered Spatial Intelligence Across Scales: Advances in Embodied Agents, Smart Cities, and Earth Science. arXiv preprint 2025.
>
> > Comment 2: As mentioned in the limitations section, the framework does not have a feedback loop between the simulator results and construction results.
> >
>
> **Reply**: We agree that closed-loop feedback plays an important role in engineering construction and optimization. To test the performance gain by integrating the closed-loop feedback, we have conducted additional experiments. We randomly select 18 samples from the failure cases of Seed-1.6 model in the Support-Lv.1 task, and add the latest machine structure and simulation results as feedback to the workflow's initial input for the next turn, achieving closed-loop engineering construction. As shown in the following Table 2, closed-loop feedback significantly improves the performance of construction outcomes, with most samples meeting test standards after the first refinement round and all passing after four more rounds. However, this improvement comes at the cost of increased inference expenses (token count).
>
> Table 2: Multi-turn closed-loop construction results of Seed-1.6 model on Support-Lv.1 task (Note: Round 1 means no closed-loop).
>
> |  | Round 1 | Round 2 | Round 3 | Round 4 | Round 5 |
> | --- | --- | --- | --- | --- | --- |
> | Success Rate (%) ↑ | 0.0 | 72.2 | 83.3 | 94.4 | 100 |
> | Indicator (Max Load) ↑ | 0.0 | 422.6 | 477.2 | 568.8 | 586.9 |
> | Number of Parts | 26.5 | 61.0 | 49.8 | 82.7 | 43.0 |
> | Invalid-Action Rate (%) ↓ | 1.2 | 1.1 | 2.2 | 0.9 | 1.2 |
> | # LLM Requests ↓ | 56.8 | 128.8 | 108.2 | 171.7 | 89.0 |
>
> Meanwhile, we are conducting ongoing experiments on Support-Lv.3 task. Sampling on this task takes much more time due to the increased task difficulty. We promise to update the results on this page within one week, and these results and corresponding analysis will also be added to the manuscript.
>
> Current experimental results in the above Table 2 show that closed-loop sampling enhances models' construction capabilities as the number of rounds increases. However, this is a benchmark paper aiming to clearly distinguish the capabilities of different LLMs. If multi-round sampling is adopted for all models, their final performance may become less distinguishable. Therefore, we adopt single-round evaluation as the default setting in our paper.

---

> > ### Author Response · Authors · 2025-11-23
> >
> > > Comment 3: The evaluation seems to conflate model capabilities with the specific 5-agent workflow and I do not know if the poor performance reflects model limitations or suboptimal orchestration. Have you tested simpler workflows? How sensitive are the results to the 5-agent setup you currently adopt?
> > >
> >
> > **Reply**: Thanks for this insightful comment. We agree that disentangled verifications of model capabilities and agentic workflows are crucial for conducting clear comparative analyses. We performed additional experiments to compare different workflows. The corresponding results are presented in Table 3 below. The results demonstrate that the performance of LLMs depends on workflow design, where for much simpler workflows, the performance would degrade significantly. Our current 5-agent workflow is derived from extensive evaluations of various workflow orchestration strategies, and this specific workflow emerges as the optimal option on average. We have supplemented these experimental results in Table 3 and analyses in Appendix I of the revised manuscript.
> >
> > Table 3: Ablation study on multi-agent pipeline for Seed-1.6 model on Support-Lv.1 task. Performance changes relative to the 5-Agent baseline are shown in parentheses.
> >
> > | Workflow | Number of Parts | Success Rate (%) ↑ | Indicator (Max Load) ↑ | Invalid-Action Rate (%) ↓ |
> > | --- | --- | --- | --- | --- |
> > | **5-Agent** | **33.4** | **45.3** | **197.4** | **1.4** |
> > | Guidance-Builder | 20.2 | 4.7(-40.6) | 31.4 (-166.0) | 1.7(+0.3) |
> > | Builder | 19.8 | 22.6 (-22.7) | 75.5 (-121.9) | 44.7(+43.3) |
> >
> > > Comment 4: How do you ensure physics consistency between your spatial geometric library and Besiege's internal engine?
> > >
> >
> > **Reply**: Our spatial geometric library follows the same construction logic as Besiege: each new block's position and orientation are computed according to the existing structure (which also matches real-world assembly rules). The geometries of blocks are extracted from Besiege's saved machine files, where all block positions and rotations are recorded.
> >
> > To verify consistency, we build the same machines both by calling functions in our library and directly inside Besiege, respectively. The parameters of all blocks in the saved machine files match exactly.
> >
> > > Comment 5: Have you tried using a higher fidelity engine (maybe MuJoCo)?
> > >
> >
> > **Reply**: We did not use other high-fidelity simulators like MuJoCo. Our paper aims to build a benchmark for testing LLM's construction capabilities, thus the simulator must meet three necessary conditions: (1) support step-by-step construction, (2) the construction process complies with physical laws, (3) the functions related to construction are open-source or can be encapsulated as API calls. Our adopted Besiege is the only one that meets these conditions. Although MuJoCo is a high-fidelity physical simulator, it is primarily used to simulate the dynamic behaviors of already constructed mechanical systems, rather than supporting interactive construction processes. Therefore, it is not suitable for our task.

---

> ### Author Response · Authors · 2025-12-03
>
> Following our previous update, we have completed the additional experiments on the high-difficulty Support-lv3 task. The results (Table 9) present a contrasting and intriguing perspective compared to the simpler tasks. Unlike Support-lv.1, where closed-loop feedback secured a 100% success rate, the Support-lv.3 samples remained at **0/18 success across all 5 turns** of refinement.
>
> Table 9: Multi-turn closed-loop construction results of Seed-1.6 model on Support-Lv.3 task (Note: Round 1 means no closed-loop).
>
> |  | Round 1 | Round 2 | Round 3 | Round 4 | Round 5 |
> | --- | --- | --- | --- | --- | --- |
> | Success Rate (%) ↑ | 0.0 | 0.0 | 0.0 | 0.0 | 0.0 |
> | Indicator (Max Load) ↑ | 0.0 | 0.0 | 0.0 | 0.0 | 0.0 |
> | Number of Parts | 98.0 | 52.6 | 66.2 | 69.9 | 58.5 |
> | Invalid-Action Rate (%) ↓ | 16.4 | 17.5 | 16.5 | 12.2 | 14.7 |
> | # LLM Requests ↓ |197.1 | 224.7 | 216.3 | 183.8 | 283.9 |
>
> We observe that for highly complex structural problems, early failure modes often narrow the decision space. The model tends to "patch" a fundamentally flawed design rather than reimagine it in a whole new perspective, effectively getting trapped in local optima. While these 18 refined seeds failed, successful designs **do exist** within our dataset when using independent random sampling. For difficult tasks, **"starting over" (broad exploration) often shines where "fixing" (deep exploitation) tends to fail.**
>
> This finding aligns with recent literature on test-time compute, such as scaling inference-time searching [10], which demonstrates that scaling simple random sampling can be a highly effective strategy.
>
> Conclusion: Closed-loop refinement is not a universal guarantee of success; its efficacy is highly sensitive to task difficulty and the quality of the initial seed. Since relying on closed-loop could mask the model's ability to generate high-quality initial candidates, and does not consistently solve hard tasks. We reaffirm that **single-round evaluation** remains the most objective and consistent metric for distinguishing inherent model capabilities in this benchmark.
>
> [10] Zhao et al. Sample, Scrutinize and Scale: Effective Inference-Time Search by Scaling Verification. arXiv preprint 2025.

---

### Official Review · Reviewer_6rvr · 2025-11-04

**Soundness:** 3
**Presentation:** 3
**Contribution:** 3
**Rating:** 6
**Confidence:** 4

**Summary:**

This paper introduces BuildArena, a physics-aligned interactive benchmark for evaluating LLMs in language-driven engineering construction. The benchmark enable models to construct and test 3D structures under physical constraints using natural language.
The paper evaluates eight major LLMs across physics-based construction tasks, reporting success rates, performance indicators, and token-cost analyses. Results show that while models exhibit rudimentary 3D construction skills and creative strategies, they still fail at compositional precision, hierarchical assembly, and spatial conflict resolution.

**Strengths:**

- BuildArena pioneers the integration of language, physics simulation, and 3D assembly for LLM benchmarking.

- The paper is well-structured, systematically progressing.

- BuildArena establishes a foundational benchmark for evaluating LLMs’ physics-grounded reasoning and interactive 3D construction.

**Weaknesses:**

- The benchmark currently performs single-round evaluation. There is no closed-loop feedback integrating simulation outcomes into iterative model improvement or self-correction.

- The 3D Spatial Geometric Computation Library, though impressive, mirrors only a subset of Besiege's physics primitives. As the authors admit, limited module diversity constrains object complexity and realism.

- The evaluation metrics capture outcome quality but miss process-level score, i.e., whether models optimize design efficiency, robustness trade-offs, or use feedback adaptively.

- Since tasks involve 3D spatial reasoning, a natural baseline would include VLMs.

**Questions:**

- Whether the paper use specific prompt across different LLMs?

- How model perform when using multi-round CoT?

- How do BuildArena tasks compare to human performance under textual-only conditions?

---

> ### Author Response · Authors · 2025-11-23
>
> We thank the reviewer for the valuable and detailed feedback. We are glad that the reviewer recognizes our work's pioneering contribution to the community and finds it well-structured and systematically progressing. Below, we address the reviewer's questions one by one.
>
> > Comment 1: The benchmark currently performs single-round evaluation. There is no closed-loop feedback integrating simulation outcomes into iterative model improvement or self-correction.
> >
>
> **Reply**: We agree that closed-loop feedback plays an important role in engineering construction and optimization. To test the performance gain by integrating the closed-loop feedback, we conduct additional experiments. We randomly select 18 samples from the failure cases of Seed-1.6 model in the Support-Lv.1 task, and add the latest machine structure and simulation results as feedback to the workflow's initial input for next turn, achieving closed-loop engineering construction. As shown in the following Table 1, closed-loop feedback significantly improves the performance of construction outcomes, with most samples meeting test standards after the first refinement round and all passing after four more rounds. However, this improvement comes at the cost of increased inference expenses (token count).
>
> Table 1: Multi-turn closed-loop construction results of Seed-1.6 model on Support-Lv.1 task (Note: Round 1 means no closed-loop).
>
> |  | Round 1 | Round 2 | Round 3 | Round 4 | Round 5 |
> | --- | --- | --- | --- | --- | --- |
> | Success Rate (%) ↑ | 0.0 | 72.2 | 83.3 | 94.4 | 100 |
> | Indicator (Max Load) ↑ | 0.0 | 422.6 | 477.2 | 568.8 | 586.9 |
> | Number of Parts | 26.5 | 61.0 | 49.8 | 82.7 | 43.0 |
> | Invalid-Action Rate (%) ↓ | 1.2 | 1.1 | 2.2 | 0.9 | 1.2 |
> | # LLM Requests ↓ | 56.8 | 128.8 | 108.2 | 171.7 | 89.0 |
>
> Meanwhile, we are conducting ongoing experiments on Support-Lv.3 task. Sampling on this task takes much more time due to the increased task difficulty. We promise to update the results on this page within one week, and these results and corresponding analysis will also be added to the manuscript.
>
> Current experimental results in the above Table 1 show that closed-loop sampling enhances models' construction capabilities as the number of rounds increases. However, this is a benchmark paper aiming to clearly distinguish the capabilities of different LLMs. If multi-round sampling is adopted for all models, their final performance may become less distinguishable. Therefore, we adopt single-round evaluation as the default setting in our paper.
>
> > Comment 2: The 3D Spatial Geometric Computation Library, though impressive, mirrors only a subset of Besiege's physics primitives. As the authors admit, limited module diversity constrains object complexity and realism.
> >
>
> **Reply**: The full set of physics primitives in Besiege is rich, but many are unsuitable for testing the construction capabilities of LLMs. These inappropriate modules mainly fall into two categories:
>
> 1. Modules with behaviors and effects violate physics, such as hot air balloons hovering at any altitude and lift fans without angular momentum. Using these modules will result in construction outcomes physically misaligned with the real world.
> 2. Modules with overly powerful functions, such as fireworks that directly enable rocket-like flight and long wooden blocks with extremely high mechanical strength. These modules allow direct shortcuts instead of step-by-step construction, bypassing the spatial reasoning of LLMs. This greatly reduces the difficulty of tasks and leads to excessively high success rates. Thus, the construction capabilities of LLMs will be overestimated.
>
> After excluding such inappropriate modules, the combinatorial space remains remarkably large for the remaining block set when the block count is unconstrained. In practice, we observe highly diverse and sometimes surprisingly complex machines (see Figure 9 in the original submission). The current module set already supports rich solution strategies.
>
> As a pioneering benchmark of its kind, we focus on establishing a controlled and clean setting rather than maximizing module diversity. The future extensions can easily include more or even self-defined components.

---

> ### Author Response · Authors · 2025-11-23
>
> > Comment 3: The evaluation metrics capture outcome quality but miss process-level score, i.e., whether models optimize design efficiency, robustness trade-offs, or use feedback adaptively.
> >
>
> **Reply**: We agree that process-level metrics are important directions. Nevertheless, our evaluation framework already logs rich intermediate information, including block-by-block actions, failure signals, and structure evolution, which can support more fine-grained analysis. We view these metrics as natural extensions rather than our core components. Future users can easily plug in their own process-level scoring rules using the recorded trajectories. We intend to provide the first stable, discriminative benchmark and a platform that enables such analyses, rather than define an endless list of evaluation dimensions in this work.
>
> > Comment 4: Since tasks involve 3D spatial reasoning, a natural baseline would include VLMs.
> >
>
> **Reply**: We agree that VLMs are a natural direction for 3D construction tasks. However, this paper aims to benchmark the 3D construction capabilities of current LLMs, which is an underexplored research area in evaluating LLMs as we analyzed in the Section of Introduction. **We do not intend to evaluate on VLMs.** Namely, we aim to isolate the model's 3D spatial reasoning ability purely from textual state descriptions. The text feedback we provide is complete and precise. Introducing visual feedback will confound the evaluation of the 3D reasoning capabilities with interference from 3D perception capabilities.
>
> > Comment 5: Whether the paper use specific prompt across different LLMs?
> >
>
> **Reply**: As a benchmark, we test every LLM model with the **exact same** designed prompt, settings, and pipeline. All LLMs see **the same** instructions and go through the same evaluation steps. Details of the prompt have been provided in **Appendices G and H in our original submission**. We consider this uniformity essential for the fairness of evaluation. And we have explicitly emphasized this standardization strategy in the revised manuscript.
>
> > Comment 6: How model perform when using multi-round CoT?
> >
>
> **Reply**: Thanks for the question. As we know, your proposed "multi-round CoT" is not standard terminology in LLM community. This term appears in only one literature [1], which actually refers to the better-known and widely used "self-consistency" [2]. It samples multiple trajectories and then selects the most common result via majority voting. The selections are based on direct **equivalence** comparison over sampling results, such as numbers in mathematical reasoning, and indices of choices in multiple-choice natural language question answering. In our task, however, it is difficult to define a metric to measure consistency between sampling results of two trajectories since each trajectory produces a 3D complex building structure. Therefore, "self-consistency" could hardly be applied to our sampling.
>
> If the reviewer would like to clarify the concept of multi-round CoT, we welcome further discussion.
>
> Reference
>
> [1] Qi et al. Mutual Reasoning Makes Smaller LLMs Stronger Problem-solvers. arXiv preprint 2024.
>
> [2] Wang et al. Self-Consistency Improves Chain of Thought Reasoning in Language Models. ICLR 2023.

---

> ### Author Response · Authors · 2025-11-23
>
> > Comment 7: How do BuildArena tasks compare to human performance under textual-only conditions?
> >
>
> **Reply**: To compare construction capabilities between humans and LLMs, we have deliberately developed a playable text-only version of BuildArena during the discussion period. It has been included in the updated anonymous **[codebase](https://anonymous.4open.science/r/BuildArena-9B7B/)**. Subsequently, an experienced human player conducted text-only construction on two tasks of different types and difficulty levels, following the same evaluation settings as in the paper. The player's performance results are shown below.
>
> - In the Transport-Lv.1 task, since the task can be easily solved with a simple 4-wheel vehicle constructed with only a few blocks, the human player can precisely complete the building process within minutes. Compared with the excellent human performance in this task, all the LLMs perform below 25% (please refer to Table 2 of the original submission).
> - In the Support-Lv.3 task, which requires highly complex bridges composed of hundreds of blocks, the human player failed to deliver a successful construction even after dozens of minutes of trying. The main challenge for the human player lies in precisely memorizing all the information of a huge 3D structure without visual feedback. All LLMs also failed except Seed-1.6. It managed to build a giant bridge to fulfill the task (See Table 2 in the original submission). This demonstrates that for complex, large-scale construction tasks, cutting-edge LLMs have the potential to surpass human capabilities.
>
> However, an accurate human vs LLM comparison requires the design of rigorous, systematic, and large-scale human tests, which goes beyond the scope of this paper.

---

> > ### Comment · Reviewer_6rvr · 2025-11-25
> >
> > Thanks for your reply, I will maintain my rating.

---

> ### Author Response · Authors · 2025-12-03
>
> Following our previous update, we have completed the additional experiments on the high-difficulty Support-lv3 task. The results (Table 8) present a contrasting and intriguing perspective compared to the simpler tasks. Unlike Support-lv.1, where closed-loop feedback secured a 100% success rate, the Support-lv.3 samples remained at **0/18 success across all 5 turns** of refinement.
>
> Table 8: Multi-turn closed-loop construction results of Seed-1.6 model on Support-Lv.3 task (Note: Round 1 means no closed-loop).
>
> |  | Round 1 | Round 2 | Round 3 | Round 4 | Round 5 |
> | --- | --- | --- | --- | --- | --- |
> | Success Rate (%) ↑ | 0.0 | 0.0 | 0.0 | 0.0 | 0.0 |
> | Indicator (Max Load) ↑ | 0.0 | 0.0 | 0.0 | 0.0 | 0.0 |
> | Number of Parts | 98.0 | 52.6 | 66.2 | 69.9 | 58.5 |
> | Invalid-Action Rate (%) ↓ | 16.4 | 17.5 | 16.5 | 12.2 | 14.7 |
> | # LLM Requests ↓ |197.1 | 224.7 | 216.3 | 183.8 | 283.9 |
>
> We observe that for highly complex structural problems, early failure modes often narrow the decision space. The model tends to "patch" a fundamentally flawed design rather than reimagine it in a whole new perspective, effectively getting trapped in local optima. While these 18 refined seeds failed, successful designs **do exist** within our dataset when using independent random sampling. For difficult tasks, **"starting over" (broad exploration) often shines where "fixing" (deep exploitation) tends to fail.**
>
> This finding aligns with recent literature on test-time compute, such as scaling inference-time searching [10], which demonstrates that scaling simple random sampling can be a highly effective strategy.
>
> Conclusion: Closed-loop refinement is not a universal guarantee of success; its efficacy is highly sensitive to task difficulty and the quality of the initial seed. Since relying on closed-loop could mask the model's ability to generate high-quality initial candidates, and does not consistently solve hard tasks. We reaffirm that **single-round evaluation** remains the most objective and consistent metric for distinguishing inherent model capabilities in this benchmark.
>
> [10] Zhao et al. Sample, Scrutinize and Scale: Effective Inference-Time Search by Scaling Verification. arXiv preprint 2025.

---

### Author Response · Authors · 2025-11-23
**General Response**

We sincerely thank reviewers for their dedicated time and constructive comments. We are pleased that **all the reviewers** recognized our work for **filling a critical gap** in evaluating LLMs' physically-grounded, multi-step 3D construction capabilities and **providing a comprehensive framework** with systematic tasks, clear metrics, and thorough empirical analysis. They also appreciated our **valuable contributions to the community**, including the foundational BuildArena benchmark (Reviewer 6rvr) and the open-source 3D Spatial Geometric Computation Library supporting future research in this underexplored area (Reviewers mYoq and TqAt).

Based on the reviewers' insightful comments, we have conducted additional experiments and made clarifications to address their concerns. In the revised manuscript, we have updated both the main text and appendices, with changes highlighted in **blue**. **The major improvements and new experiments are summarized as follows:**

1. We add a closed-loop study where the agents iteratively use simulation feedback to refine their designs, in response to Review 6rvr, mYoq, fcZf, and TqAt. The results are presented in the following Table 1, showing that the outer-loop significantly improves the performance of construction outcomes after refinement. For more details, please refer to responses to Reviewer 6rvr, mYoq, fcZf, and TqAt. The experiment results are provided in Table 6 in the revised manuscript.
2. We clarify the physics-aligned benchmark for modular engineering construction as the core contributions as suggested by Reviewer fcZf. The other technical elements are developed to ensure the stability and differentiation ability of the benchmark. We make modifications accordingly in the Abstract and Introduction Section in the revised manuscript. More details can be found in the responses to Reviewer fcZf.
3. We conduct comprehensive ablations over (i) workflow design (e.g., single-LLM ReAct, Plan-Execute), (ii) prompt formats (zero-shot versus few-shot), and (iii) decoding parameters, in response to Reviewer mYoq and fcZf. These experiments demonstrate that the proposed pipeline and library components are necessary and beneficial for achieving robust and discriminative performance. See responses to Reviewer mYoq and fcZf for more details. The experiment results are updated in Table 4, and Table 5 in the revised manuscript.
4. We provide a deeper qualitative analysis of how and why LLMs produce spatial conflicts as suggested by Reviewer mYoq and TqAt. These failures originated from fundamental limitations of current LLMs, can be suppressed by our designed workflow and Spatial Computation Library, but can not be eliminated. See response to Reviewer mYoq and TqAt for more details.
5. We develop an interface of textual-only building processes for introducing human participants to BuildArena tasks, following the suggestion of Reviewer 6rvr. Comparison between human players and LLMs reveals the potential advantages of LLMs in large-scale complex tasks. More details can be found in the responses to Reviewer 6rvr.

We believe these revisions significantly strengthen the paper and directly resolve the main concerns raised by the reviewers.

---

### Author Response · Authors · 2025-12-03
**Summary of the Discussion Period**

Dear Area Chair,

Thank you for your time and effort in the evaluation of our submission. In light of the Area Chair reassignment and the closing of the discussion phase, we wish to present a brief summary of the progress already made during the rebuttal and declare the fundamental contributions of our research.

During the rebuttal, we actively engaged with the reviewers to resolve their inquiries, resulting in a score increase from 2 to 4 by Reviewer TqAt, while Reviewer 6rvr reaffirmed their positive assessment of 6. Although Reviewers mYoq and fcZf have not responded to our updates, we have thoroughly addressed their concerns regarding the benchmark's robustness. Specifically, we conducted and reported additional experiments covering various agentic workflows, LLM decoding settings, and closed-loop refinement using simulation feedback. These supplementary results demonstrate the validity and discriminative power of BuildArena as an LLM benchmark. We believe our detailed responses and new experimental evidence strongly support the merits of our submission and effectively answer the pending concerns.

The reviewers have uniformly recognized BuildArena's contribution as the **first physics-aligned interactive benchmark** for engineering construction. By bridging the gap between high-level semantic planning and low-level physical execution, our work **pioneers the integration of language, physics simulation, and 3D assembly**. Key contributions acknowledged by the reviewers include the **systematic design of task families** that cover distinct engineering dimensions, and the development of an open-source **3D Spatial Geometric Computation Library**. This library is highlighted as a significant engineering contribution that enables the research community to perform **grounded spatial reasoning** previously inaccessible in standard text-to-code benchmarks.

Beyond its novelty, BuildArena serves as a **foundational benchmark** for the agent and LLM communities, addressing a clear gap in evaluating physically grounded reasoning. Our evaluation **reveals critical limitations in frontier models**, specifically their struggles with **precise 3D spatial reasoning and construction** despite having semantic understanding. As noted by the reviewers, our analysis of failure modes and token costs **surfaces stable weaknesses** in current LLMs, proving that larger context windows alone do not solve physical grounding. By providing a **comprehensive, extendable testbed**, BuildArena offers the community a vital platform to investigate these defacts and advance the reliability of **LLM agents in the physical world**.

Best regards,

Authors of ICLR Submission 2752

---

### Note · Program_Chairs · 2026-01-17
**Submission Desk Rejected by Program Chairs**

The following references in this submission do not refer to real documents and/or have major errors in bibliographic information:

 Abhishek Kamble, Dnyaneshwar Gadekar, and Sopan Kadam. The role of precision metrology in enhancing manufacturing quality: A comprehensive review. International Research Journal of Modernization in Engineering Technology and Science, 6(2):2788-2794, 2024. URL https://www.researchgate.net/publication/378864357_THE_ROLE_ OF_PRECISION_METROLOGY_IN_ENHANCING_MANUFACTURING_QUALITY_A_ COMPREHENSIVE_REVI